

# Estimating nitrogen and sulfur deposition across China during 2005-2020 based on multiple statistical models

Kaiyue Zhou[1], Wen Xu[3], Lin Zhang[4], Mingrui Ma[1], Xuejun Liu[3], Yu Zhao[1,2*]
1. State Key Laboratory of Pollution Control & Resource Reuse and School of the
Environment, Nanjing University, Nanjing, Jiangsu 210023, China
2. Jiangsu Collaborative Innovation Center of Atmospheric Environment and
Equipment Technology (CICAEET), Nanjing University of Information Science &
Technology, Nanjing, Jiangsu 210044, China
3. Key Laboratory of Plant-Soil Interactions of MOE, College of Resources and
Environmental Sciences, National Academy of Agriculture Green Development, China
Agricultural University, Beijing 100193, China
4. Laboratory for Climate and Ocean-Atmosphere Sciences, Department of
Atmospheric and Oceanic Sciences, School of Physics, Peking University, Beijing
100871, China
*Corresponding author: Yu Zhao
Phone: 86-25-89680650; email: yuzhao@nju.edu.cn



## Abstract

Due to the rapid development of industrialization and substantial economy, China has become one of the global hotspots of nitrogen (N) and sulfur (S) deposition following Europe and the USA. Here, we developed a dataset with full coverage of N and S deposition from 2005 to 2020, with multiple statistical models that combine ground-level observations, chemistry transport simulations, satellite-derived vertical columns, and meteorological and geographic variables. Based on the newly developed random forest method, the multi-year averages of dry deposition of OXN, RDN and S in China were estimated at 10.4, 14.4 and 16.7 kg N/S ha$^{-1}$ yr$^{-1}$, and the analogous numbers for total deposition were respectively 15.2, 20.2 and 25.9 kg N/S ha$^{-1}$ yr$^{-1}$ when wet deposition estimated previously with a GAM model was included. The $R_{\text{dry/wet}}$ of N stabilized in earlier years and then gradually increased especially for RDN, while that of S declined for over ten years and then slightly increased. $R_{\text{RDN/OXN}}$ was estimated to be larger than 1 for the whole research period and clearly larger than that of the USA and Europe, with a continuous decline from 2005 to 2011 and a more prominent rebound afterwards. Compared with the USA and Europe, a more prominent lagging response of OXN and S deposition to precursor emission abatement was found in China. The OXN dry deposition presented a descending gradient from east to west, while the S dry deposition a descending gradient from north to south. After 2012, the OXN and S deposition in eastern China declined faster than the west, attributable to stricter emission controls. Positive correlation was found between regional deposition and emissions, while smaller deposition to emission ratios (D/E) existed in developed eastern China with more intensive human activities.

## 1. Introduction

Atmospheric deposition of nitrogen (N) and sulfur (S) is considered as a serious environmental problem, leading to widespread ecosystem acidification and eutrophication, as well as human health damages (Baker et al., 1991; Burns et al., 2016;



Payne et al., 2011; Reuss et al., 1987; Zhang et al., 2018a). In order to understand the
spatial distribution and temporal variability of deposition, long-term observation
networks have been established globally particularly in developed countries or regions,
such as Clean Air Status and Trends Network/the National Atmospheric Deposition
Program (CASTNET/NADP) in the USA (Beachley et al., 2016), Canadian Air and
Precipitation Monitoring Network (CAPMoN) in Canada (Cheng et al., 2022),
European Monitoring and Evaluation Program (EMEP) in Europe (Simpson et al.,
2012), and Acid Deposit Monitoring Network in East Asia (EANET; Tørseth et al.,
2012; Totsuka et al., 2005; Yamaga et al., 2021). Reductions of anthropogenic $NO_X$ and
$SO_2$ emissions in North America have been very effective in reducing the oxidized
nitrogen (OXN) and wet S deposition (Cheng and Zhang, 2017; Feng et al., 2021;
Likens et al., 2021). In the USA, for example, OXN decreased significantly in most
areas, while reduced nitrogen (RDN) increased gradually in agricultural areas (Holland
et al., 2005; Li et al., 2016). Similarly, the long-term observation in Europe shows a
downward trend for N and S deposition over the last two decades (Keresztesi et al.,
2019; Theobald et al., 2019).
China has become one of global hotspots of atmospheric deposition due mainly to
the large anthropogenic emissions from increased industrial economy and energy
consumption for the past two decades (Vet et al., 2014). To reduce soil acidification
and improve air quality, the Chinese government has enacted a series of policies to cut
the emissions of atmospheric deposition precursors since 2005 (Li et al., 2017; Liu et
al., 2015; Zheng et al., 2018a), including the policy of limiting national total emission
levels of $SO_2$ and $NO_X$ within the 11[th] Five-year Plan (FYP) period (2005-2010), the
National Action Plan on the Prevention and Control of Air Pollution (NAPPCAP,
2013-2017), and the Three-Year Action Plan to fight air pollution (TYAPFAP,
2018-2020). Estimated by the Multiple-resolution Emission Inventory for China
(MEIC, http://www.meicmodel.org), those policies have reduced annual $SO_2$ and $NO_X$
emissions from different years (Li, 2020; Wang et al., 2022; Zhang et al., 2019), while



the change in $NH_3$ was relatively small. The $SO_2$ and $NO_X$ vertical column densities
(VCDs) measured from satellite remote sensing have also declined to varying degrees
across the country (Krotkov et al., 2016; Xia et al., 2016). Besides emissions and
ambient columns, accurate estimation on the changing N and S deposition is crucial for
evaluating the effectiveness of national policies on decreasing the ecological risk.
Limited by data and methods (explained below), however, few studies have been
conducted to link the long-term trend of deposition to the regulations of air pollution
prevention.

Similar to developed countries, the direct knowledge of deposition in China came

first from ground observation. Since 1990s, atmospheric deposition monitoring
networks in China have been gradually established and improved, such as the Chinese
Nationwide Nitrogen Deposition Monitoring Network (NNDMN; Xu et al., 2019) and
the Chinese Ecosystem Research Network (CERN; Fu et al., 2010). They provide
essential information for quantifying dry and wet deposition and revealing its
long-term variability at site level. For example, Liu et al. (2013) found a significant
growth in bulk nitrogen deposition in China between 1980 and 2010 based on
meta-analyses of historical observation data. Due to insufficient spatial and temporal
coverage, however, data obtained at individual sites could not fully support the analysis
of widespread and long-term evolution of deposition and might miss diverse patterns
of changing deposition by region (Hou et al., 2019; Lye and Tian, 2007). Statistical
methods, which incorporated meteorological and environmental variables with higher
temporal and horizontal resolutions and wide coverage in time and space (e.g.,
satellite-derived VCDs), have been increasingly applied to fill the observation gap.
Linear or nonlinear relationship between those variables and observed deposition have
been developed and applied for periods and regions without observation (Jia et al.,
2016; Xu et al., 2018; Yu et al., 2019). For example, Liu et al. (2017a) and Zhang et al.
(2018b) obtained the removal rate of $SO_2$ and $NO_X$ by precipitation in the whole
atmospheric boundary layer through linear regression method, and estimated the wet S



deposition in 2005-2016 and nitrogen in 2010-2012 in China. Relatively high
uncertainty existed in the simple linear assumption, given the complicated effects of
multiple variables (e.g., meteorological conditions and underlying surface types) on
deposition. Although advanced statistical methods such as k-Nearest Neighbor (KNN),
Gradient Boosting Machine (GBM) and neural networks have been developed to
predict the air pollutant concentrations, they are much rarely used in the estimation of
deposition (Li et al., 2020b; Li et al., 2019; Qin et al., 2020; Wu et al., 2021). Out of
the limited studies, Li et al. (2020a) developed machine learning prediction methods
based on multi-sites observation data and integrated meteorological and land use type
information, which improved the prediction accuracy of temporal and spatial
distribution of ammonium wet deposition.

Besides spatiotemporal coverage, integrated estimation for multiple species is

another great challenge, particularly for dry deposition. Compared with wet or bulk
deposition, there are very few data available for direct observation of dry deposition
and an "inferential method" that incorporates numerical-simulated dry deposition
velocity ($V_d$) and surface concentration has been commonly applied (Cheng et al., 2012;
Luo et al., 2016; Wesely, 1989; Xu et al., 2015; Wen et al., 2020). Notably, there are
even fewer studies on the dry deposition of secondary-formation species with neither
surface nor satellite observation data available at the regional scale (e.g., nitrate,
ammonium, and sulfate). Chemistry transport modeling (CTM), which takes
mechanisms of secondary formation of atmospheric species into account, is able to
provide the temporal and spatial distribution of ambient concentration of those species,
thus can potentially be incorporated into the machine learning framework to improve
the deposition estimation and complete the information for individual species. Such
application (combination of CTM and machine learning in deposition estimation) has
been seldom reported to our knowledge.

In response to the above limitations, this study aims to develop a machine

learning framework for estimating the historical long-term deposition of multiple N



and S species at relatively high horizontal (0.25°×0.25°) and temporal resolution
(monthly) for China, and to explore the comprehensive impact of the national air
pollution controls on the deposition. We select the period 2005-2020, which covers
three national FYP periods (11[th]-13[th]), NAPPCAP and TYAPFAP. We applied a random
forest (RF) method and a generalized additive model (GAM combining different
datasets, including ground-level deposition observation, satellite-derived VCDs,
meteorological and geographic variables, and CTM simulation, and explore the
spatiotemporal variability of dry and wet deposition for the country. The ratios of
deposition to emissions ($D/E$) were then calculated by region and species to illustrate
the source–sink relationships of atmospheric pollutants. The outcomes provide
scientific basis for further formulating emission control strategies, combining potential
ecological risks of deposition.
**2. Materials and methods**
**2.1 Study domain**
We selected Chinese mainland as the research area including 31 provincial-level
administrative regions (excluding Hong Kong, Macao and Taiwan). As shown in
Figure 1, the 31 provinces are geographically classified into 6 parts, i.e., North Central
(NC), North East (NE), North West (NW), South East (SE), South West (SW), and the
Tibetan Plateau (TP), representing the diverse social-economical and geo-climatic
conditions. The details in climate, population and GDP are provided by region in Table
S1 in the Supplement. Basically, NC (with Inner Mongolia excluded) and SE belong to
the relatively developed regions in eastern China, NW, SW and NE belong to less
developed regions, while TP represents the background region. Bounded by the
Qinling Mountain-Huaihe River Line (Figure 1), the climate in the south (SE and SW)
is humid with more precipitation than the north (e.g., NC).





**2.2 Dry deposition flux estimation**

**2.2.1 Random forest (RF) model description**

Figure 2 shows the methodology framework of dry and wet deposition simulation. We applied a multisource-fusion RF model to estimate the spatiotemporal pattern of dry deposition for individual N and S species including $NO_3^-$, $HNO_3$, $NO_2$, $NH_4^+$, $NH_3$, $SO_2$, and $SO_4^{2-}$ ($H_2SO_4$ is not included due to its tiny amount and unavailability of relevant data), at 0.25°×0.25° horizontal resolution and monthly level for 2005-2020. RF model is a state-of-art statistical method to deal with the complicated nonlinear relationship between response variable and interpretation variables. Briefly, with the ensemble learning, the RF regression predictions are determined as the average of the multiple regression trees based on the bootstrap sampling method (Breiman, 2001). The model performance strongly depends on two crucial parameters, *ntree* (number of the regression trees) and *mtry* (number of interpretation variables sampled for splitting at each node), and they were respectively determined at 1000 and 3 to train our model. Not all interpretation variables participate in the process of node splitting (Li et al., 2020b), thus significant correlations of regression trees can be avoided. Besides, the backward variable selection was performed on the RF model to achieve the better performance. Please refer to SI Text Section for the detailed algorithm of the model.

We ran the RF modeling program by using the "caret" package in R software (version 4.1.2; Kuhn, 2021). As shown in Figure 2, we firstly selected satellite-derived tropospheric vertical columns densities (VCDs), meteorological factors, geographic covariates and chemical transport mode (CTM) results as interpretation variables, and calculated the dry deposition flux ($F_d$) at ground observation sites as response variable:

$$F_d = C \times V_d \tag{1}$$

where $C$ is the estimated (for $SO_4^{2-}$) or observed concentration (for other species) described in Section 2.2.2, and $V_d$ is the modeled dry deposition rates ($V_d$) with the



Goddard Earth Observation System-Chemistry (GEOS-Chem) 3-D global transport
model (http://geos-ch em.org) described in Section 2.2.4.

Secondly, we used the "nearZeroVar" function in "caret" package to eliminate the

zero variance variables, to delete highly correlated variables, and to prevent the
multicollinearity. Based on the Recursive Feature Elimination (RFE), we then input the
final features to the model as summarized in Table S2 in the supplement. The RFE
algorithm is a backward selection of features based on the relative importance of
interpretation variables (RIV). In order to eliminate the different distributions/ranges
caused by the magnitudes of various variables, we mapped them to the same interval
through standardization and normalization. Before modeling, the interpretation
variables were sorted, and the less important factors were eliminated in turn. Finally,
we split the entire model fitting dataset into 10 groups to test the robustness of RF
model (10-fold cross validation). In each round of cross validation, the samples in 9
groups were used as the training data, and the remaining group was applied for
prediction. This process repeated 10 times and every group was tested. The consistency
between the calculated $F_d$ (as an observation) and predictions was evaluated using
statistical indicators, including coefficient of determination ($R^2$), root mean squared
prediction error (RMSE), mean prediction error (MPE) and relative prediction error
(RPE).
**2.2.2 Ground-level concentration observations and prediction**

The daily ground-level concentrations of $NO_2$ and $SO_2$ during 2013-2020 were

obtained from the real-time data publishing system of the China National
Environmental          Monitoring          Centre          (CNEMC,
http://datacenter.mee.gov.cn/websjzx/queryIndex.vm), with the abnormal values
eliminated. The total number of observation sites reached 1532 in 2020, mainly located
eastern China with dense industrial economic and population (e.g., 600 and 408 sites in
SE and NC, respectively), as shown in Figure 1. Monthly-level concentrations were



then calculated for RF model prediction. The Nationwide Nitrogen Deposition
Monitoring Network (NNDMN) established by China Agricultural University contains
43 monitoring sites in China (as shown in Figure 1) and measured monthly
concentrations gaseous $NH_3$, $NO_2$, and $HNO_3$ and particulate $NH_4^+$ and $NO_3^-$ in air, as
well as wet/bulk deposition from 2010 to 2014. The complete datasets of NNDMN
were published in previous work (Xu et al., 2019).
Due to the lack of large-scale ground observation data, sulfate ($SO_4^{2-}$)
concentration must be obtained with an indirect method. Given the significant positive
correlation between the two (Luo et al., 2016), we estimated a simple linear
relationship between $SO_2$ and sulfate concentration with CTM and calculated the
sulfate concentrations ($G_{SO_4^{2-}}$):
$$G_{SO_4^{2-}} = G_{SO_2} \times f(G_{CTM-SO_4^{2-}}, G_{CTM-SO_2}) \qquad (2)$$
where $G_{SO_2}$ is the monthly ground-level concentration at CNEMC; $G_{CTM-SO_4^{2-}}$,
$G_{CTM-SO_2}$ are the sulfate and $SO_2$ concentrations simulated by CTM, respectively, and
*f* is the ratio of simulated sulfate to $SO_2$ (see Section 2.2.4 for CTM description).
**2.2.3 Satellite-derived VCDs**
The tropospheric VCDs of $NO_2$ from 2005 to 2020 were taken from Peking
University OMI NO2 tropospheric product version2 (POMINO v2; Liu et al., 2019),
based on the observation of Ozone Monitoring Instrument (OMI). The VCDs with
cloud coverage over 25% were eliminated as high cloudiness would distort satellite
detection and increase inversion error. The daily $SO_2$ VCDs were obtained from
Level-3e OMSO2 Data Products from 2005 to 2020
(https://disc.gsfc.nasa.gov/datasets/OMSO2e_003/summary). All the OMI $SO_2$ data
were generated by an algorithm based on principal component analysis (PCA), which
was considerably sensitive to anthropogenic emissions (Krotkov et al., 2016). The total



VCDs of $NH_3$ were derived from the Infrared Atmospheric Sounding Interferometer
(IASI), board on MetOp-A platform. The standard daily IASI/Metop-A ULB-LATMOS
total column Level-2 product v2.2.0 is available from 2008 to 2020
(https://iasi.aeris-data.fr/nh3_iasi_a_arch/). The daily total column was excluded when
the cloud coverage was >25%, the relative error was >100%, or the absolute error
was >$5\times10^{15}$ molecules $cm^{-2}$ (Whitburn et al., 2016). The $NH_3$ VCDs from 2005 to
2008 were estimated based on the linear correlations between $NH_3$ emission and VCDs
during 2008-2020.
We used the Kriging interpolation method to fill the missing values, and obtained
the spatial pattern of VCDs at the horizontal resolution of 0.25°×0.25°. Monthly-level
VCDs were calculated based on the daily products from 2005 to 2020.
**2.2.4 CTM model description**
We used GEOS-Chem v12.1.1 to simulate the $V_d$ and the ground-level
concentrations of individual species. A nested version was applied with the native
horizontal resolution of 0.5°×0.625° over East Asia (70-150°E, 11°S-55°N) and 4°×5°
for rest of the world, and the simulated $V_d$ and concentrations within China were
spatially interpolated at the resolution of 0.25°×0.25°. As described in Section 2.2.1,
the $V_d$ for 2013-2018 was calculated based on a standard big-leaf resistance-in-series
parameterization (Wesely, 1989), and applied in estimation of the response variable dry
deposition flux. The simulated concentrations of individual species since 2005 were
used as the interpretation variable in RF.
The model was driven by the MERRA-2 assimilated meteorological data provided
by the Global Modeling and Assimilation Office (GMAO) at the National Aeronautics
and Space Administration (NASA). Meteorology fields such as vertical pressure
velocity, temperature, surface pressure, relative and specific humidity had a temporal
resolution of 3 h, and surface variables (such as sea level pressure, tropopause pressure)



and mixing depths were at 1 h resolution. The model had 47 vertical layers from
surface to 0.01 hPa, and the lowest layer is centered at 58 m above sea level.
Emissions in GEOS-Chem were processed through Harvard–NASA Emission
Component (HEMCO; Keller et al., 2014). We used the Community Emissions Data
System for global anthropogenic emissions, overwritten by the regional emissions
inventories in the USA, Europe, Canada and Asia, involving the National Emissions
Inventory     from     EPA     (NEI;
https://www.epa.gov/air-emissions-inventories/air-pollutant-emissionstrends-data),
European Monitoring and Evaluation Programme emissions (EMEP; European
Monitoring and Evaluation Programme; www.emep.int/index.html) and the MIX
inventory that included MEIC over China. Natural $NO_X$ sources from soil and
lightning were also included (Lu et al., 2021).
**2.2.5 Other data**
The meteorological parameters for 2005-2020, including precipitation, boundary
layer height, temperature at two meters, wind speed, wind direction, surface pressure,
total column, total column ozone, were downloaded from the European Centre for
Medium-Range     Weather     Forecasts     (ECMWF,
https://apps.ecmwf.int/datasets/data/interim-full-daily/levtype=sfc/) at the resolution of
0.25°× 0.25°.
Land-Use and Land-Cover Change (LUCC), Digital Elevation Model (DEM),
population density data (POP) and Gross Domestic Product (GDP) were obtained from
Chinese Resource and Environment Data Cloud Platform (http://www.resdc.cn/).
Except for the DEM, other data were compiled at a five-year interval (2005, 2010 and
2015 for this study). LUCC was generated by manual visual interpretation of Landsat
TM/ETM remote sensing image. We calculated the area fractions of different land use
in the buffer zone (60 km in diameter around each site). The elevation spatial



distribution data (DEM) were extracted from the Shuttle Radar Topography Mission at
the 1-km resolution, assuming no variability during the study period. For GDP and
POP, datasets with 1-km resolution were developed through spatial interpolation,
taking their spatial interactions with land use type and night light brightness into
account (Xu, 2017). Linear interpolation was applied to complete the information for
all the years within the research period, and all the above-mentioned interpretation
variables were resampled to a uniform horizontal resolution of 0.25°×0.25°.
**2.3 Wet deposition flux estimation**
As shown in Figure 2, we applied a nonlinear Generalized Additive Model (GAM)
developed in our previous work (Zhao et al., 2022) to estimate the monthly wet
deposition of $SO_4^{2-}$, $NO_3^-$ and $NH_4^+$ in China at a horizontal resolution of 0.25°×0.25°:
$$g(\mu_m) = \sum_{i=1}^{n} f_i(x_{i,m}) + \sum_{p,q} f_{pq}(x_{p,m}, x_{q,m}) + X_m\theta + \varepsilon_m \qquad (3)$$
where $g$ is the "link" function, which specifies the relationship between the response
variable $\mu$ and the linear formulation on the right side of equation; $f_i(x_i)$ is the nonlinear
smooth function that explores the single effect of individual interpretation variable $x_i$;
$m$ indicates the month; $n$ represents the total number of interpretation variables for
which single effect was considered in the model; $f_{pq}(x_p, x_q)$ is nonlinear smooth
function that explores the interaction effect of interpretation variable $x_p$ and $x_q$; $X\theta$
represents an ordinary linear model component for interpretation variables (elements of
the vector $X$) not subject to nonlinear transformations; and $\varepsilon$ represents the residuals of
models. The smooth functions $f_i(x_i)$ and $f_{pq}(x_p, x_q)$ are fitted by thin-plate regression
splines and tensor product smoothing, respectively. With an assumption of normal
distribution, Gaussian distribution and the log link function are applied for the model
residuals.
For $SO_4^{2-}$, the observation data of monthly wet deposition were collected from the





East Asia Acid Deposition Monitoring Network (EANET) as response variables. For
$NO_3^-$ and $NH_4^+$, the observed monthly wet or bulk deposition at NNDMN served as the
response variables. For all the three species, the interpretation variables contained the
precipitation, satellite-derived VCDs, $PM_{2.5}$ concentrations, total column liquid water,
temperature, boundary layer height, forest-cover and urban-cover. The data sources and
model performance evaluation was described in Zhao et al. (2022). Although bulk
deposition includes a small amount of dry deposition, the deposition in precipitation
obtained through GAM was uniformly defined as wet deposition in this work.

## 3. Results and discussions


### 3.1 RF model prediction performance


The RF model performances for dry deposition estimation evaluated with 10-fold
cross validation are shown in Figures S1 and S2 in the supplement based on CNEMC
and NNDMN, respectively. The multi-year average $R^2$ of N and S species over China
were all above 0.7 and the RMSE of all models were less than 1 kg N/S ha$^{-1}$ yr$^{-1}$
except for $NO_2$ (1.09 kg N ha$^{-1}$ yr$^{-1}$) and $SO_2$ (6.46 kg S ha$^{-1}$ yr$^{-1}$), indicating the
satisfying consistency between observation and prediction. However, the model tended
to underestimate the high deposition and overestimate the low one possibly because the
model algorithm based on the average of all regression trees resulted in relatively weak
estimation of the extreme values. The modeling prediction performance of OXN ($NO_3^-$,
$HNO_3$ and $NO_2$) was clearly better than that of RDN ($NH_4^+$ and $NH_3$) and sulfur ($SO_2$
and $SO_4^{2-}$). For example, the $R^2$ of $NO_2$, $NO_3^-$ and $HNO_3$ were 0.87, 0.73 and 0.78,
while those of $NH_3$ and $NH_4^+$ were only 0.71 and 0.65. POMINO, which reduced the
bias of the default product by the OMI Nitrogen Dioxide Algorithm Team (Krotkov et
al., 2019; Liu et al., 2019), was demonstrated to be satisfyingly applicable in OXN
deposition prediction for China. In addition, the prediction performances of CNEMC
were better than those of NNDMN (except for $SO_2$), attributed partly to much more





monitoring stations for the former. As indicated in our previous work, improved model
performance could be expected along with the increased abundance of observation data
(Zhou et al., 2021).
To evaluate the long-term average deposition from RF modeling, we collected 34
studies that quantified the deposition of different species and forms (dry or wet) for
China using observational, geostatistical or modal methods (Table S3 in the
supplement). As shown in Figure 3, gaseous $NH_3$ and $SO_2$ were identified as the
species with largest dry deposition, while sulfate as the species with the largest wet
deposition. The multi-year averages (2005-2020) of dry deposition for different species
estimated in this study were within the range between 25th Quantile (Q1) and 75th
Quantile (Q3) of selected studies except for $NH_3$ (Figure 3a), but that of sulfate wet
deposition closing to Q1 was basically lower compared to existing studies (Figure 3b).
Most of the existing studies reported sulfate wet deposition in China for 2001-2005
when the national control of $SO_2$ emissions and acid rain was still in its initial stage,
while limited data was available for more recent years when sharp declines were found
for $SO_2$ emissions. Therefore, the average of existing studies might potentially
overestimate the actual average level of S deposition across the country. Overall, the
total deposition of N and S from RF modeling was satisfyingly closed to the median
level of the existing studies (Figure 3c), indicating the robustness of deposition
estimation.
We calculated the shares of different forms and species to the average of national
total deposition in 2005-2020 (Figure 4). The dry deposition of N followed an order of
$NH_3 > HNO_3 > NO_2 > NH_4^+ > NO_3^-$, while the wet $NH_4^+$ deposition was larger than $NO_3^-$.
As a whole, RDN (58%) was found to contribute more than OXN (42%) to the total N
deposition. For S species, the dry deposition of $SO_2$ was over ten times of $SO_4^{2-}$, while
the latter was only species of wet deposition. Dry deposition was estimated to be
higher than wet for both N and S, with its fraction reaching 70% and 65% within the



research period, respectively. The more specific interannual variability and spatial
distribution for different forms will be described in Sections 3.2 and 3.3.
**3.2 Temporal variability in deposition of Nr species and sulfur**
Based on the newly developed RF method, the average dry deposition of OXN,
RDN, total N and S in China were estimated at 10.4, 14.4, 24.9 and 16.7 kg N/S ha$^{-1}$
yr$^{-1}$ from 2005 to 2020, respectively. The total deposition reached 15.2, 20.2, 35.4 and
25.9 kg N/S ha$^{-1}$ yr$^{-1}$, respectively, when the average wet deposition estimated with
GAM (Zhao et al., 2022) was included. Figure 5a-d illustrates the long-term
interannual variability of dry and wet deposition for OXN, RDN, total N and S,
respectively. Different temporal trends are found for N and S, due partly to the diverse
of their precursor emissions. As indicated by MEIC, China's NO$_X$ emission control was
limited before 2012, allowing annual national emissions to grow 49% from 2005 to
2012 (Figure 5f). Starting in 2013, NAPPCAP drove fast growing penetration of
selective catalyst reduction (SCR) systems in the power and cement production sectors,
resulting in a 28.6% reduction in the annual total emissions of NO$_X$ from 2013 to 2020
(Karplus et al., 2018; Li et al., 2018). Similar temporal variability was found for OXN
deposition: it was increasing slightly from 14.7 in 2005 to 15.7 kg N ha$^{-1}$ yr$^{-1}$ in 2012,
and then declining to 14.5 kg N ha$^{-1}$ yr$^{-1}$ in 2020 (Figure 5a). The interannual variation
in NH$_3$ emissions has been much smaller than NO$_X$, with a slight reduction by 9% from
2005 to 2020 (Figure 5f), attributed to the changes in Chinese agricultural practices,
e.g., improved waste management in livestock farming and replacement of highly
volatile ammonium bicarbonate with urea in fertilizer types (Liu et al., 2017b; Zheng et
al., 2018b). However, the big emission abatement of acidic gases like SO$_2$ after 2013
was recognized to reduce the sink of NH$_3$ in the atmosphere and to increase of
gas-phase NH$_3$ concentrations (Liu et al., 2018), resulting in more dry NH$_3$ deposition
(Figure 5b). After 2015, China's RDN deposition became relatively stable, which could
be partly explained by the implementation of Zero Increase Action Plan for N fertilizer





after 2015 (Liu et al., 2022). As a combined effect of changing emissions and
atmospheric conditions, the RDN deposition was estimated to grow from 19.5 in 2005
to 20.6 kg N $ha^{-1}$ $yr^{-1}$ in 2020. China has widely applied flue gas sulfurization (FGD)
in the power sector since 2005, and has expanded its application to other industries
(such as sintering furnaces and non-electric coal-fired boilers) since 2013, as a part of
NAPPCAP (Zheng et al., 2018a). As a result, the annual national $SO_2$ emissions were
estimated to decline by 76% from 2005 to 2020 (Figure 5f), and the dry deposition of S
by 31% (Figure 5d). The wet deposition was less responsive to emissions than dry
deposition, and the growth in precipitation was likely offsetting part of the benefit of
emission control on wet deposition (Zhao et al., 2022). The total S deposition was
calculated to decline 26%, from 28.8 in 2005 to 21.3 kg S $ha^{-1}$ $yr^{-1}$ in 2020.

Shown in Figure 5a-d as well is the long-term interannual variability of the dry to

wet deposition ratio ($R_{dry/wet}$) during 2005-2020. The $R_{dry/wet}$ of N species kept
relatively stable for earlier years and then gradually increased since 2015, with the
multi-year average ratios estimated at 2.2, 2.5 and 2.4 for OXN, RDN and total N,
respectively. The $R_{dry/wet}$ of sulfur declined before 2015 and then slightly increased
afterwards, with the average ratio estimated at 1.8 for 2005-2020. The growth of
$R_{dry/wet}$ of RDN could be partly attributed to the improved control of acid precursor
emissions for recent years. Since 2013, as mentioned above, implementation of
NAPPCAP and abatement of $SO_2$ emissions has reduced the sink of $NH_3$ in the
atmosphere, elevating the free ammonia in the air and thereby $R_{dry/wet}$ of RDN.
Significant negative correlation coefficient between precipitation and $R_{dry/wet}$ was found
for both OXN (-0.63) and S (-0.64), indicating the influence of precipitation. Notably,
precipitation increased at a rate of 6.3 mm $yr^{-1}$ in China during 2005-2015 (Figure S3
in the supplement), motivating the formation of wet deposition of $SO_2$ that is easily
soluble in water. The declining precipitation after 2015 resulted in the reduced wet
deposition and thereby enhanced $R_{dry/wet}$ for OXN and S. In addition, the increased




temperature after 2012 (Figure S3) could strengthen the atmospheric diffusion and the
opening of stomata of plant leave, which in turn resulted in more pollutants being
removed via dry deposition (Zhang et al., 2004).

Figure 5e shows the long-term interannual variability of the ratio of RDN to OXN

deposition ($R_{RDN/OXN}$) for different forms during 2005-2020. $R_{RDN/OXN}$ indicates the
relative contributions of industrial and agricultural activities to N deposition, as the
major anthropogenic sources of RDN are animal excrement and fertilizer use in
agriculture while those of OXN are fossil fuel combustion in power, industrial and
transportation sectors (Pan et al., 2012; Zhan et al., 2015; Zhu et al., 2015). $R_{RDN/OXN}$ is
estimated to be larger than 1 for the whole research period, with a continuous decline
from 2005 to 2011 and more prominent rebound afterwards, and it reached 1.5 for total
N in 2020. The ratio for dry deposition was larger than the wet one. The declining
$R_{RDN/OXN}$ resulted mainly from the growth of $NO_X$ emissions and thereby OXN
deposition, driven by the fast development of industrial economy and increasing fossil
fuel combustion. The growing $R_{RDN/OXN}$ since 2012 was expected to be largely driven
by the continuous efforts of $NO_X$ emission controls, and highlighted the benefit of
those efforts on limiting OXN pollution. Regulation on $NH_3$ emission controls, mainly
in agricultural activities, became increasingly important for further alleviating the N
pollution.

As summarized in Table S4 in the supplement, the annual average deposition of N

and S in China was much larger than that for USA estimated by Clean Air Status and
Trends Network (CASTNET, https://www.epa.gov/castnet) and National Atmospheric
Deposition                              Programme                              (NADP,
https://nadp.slh.wisc.edu/networks/national-trends-network/) and Europe by European
Monitoring and Evaluation Programme (EMEP, https://projects.nilu.no/ccc/index.html).
According to Vet et al. (2014), the ensemble-mean results of 21 global CTMs indicated
that eastern China was the region with the highest nitrogen deposition in the world,





with a value of 38.6 kg N ha$^{-1}$ yr$^{-1}$. Compared with USA and Europe, China has not
only experienced high deposition of N and S but also featured the greatest increase
over the past decade (Du and Liu, 2014; Fu et al., 2022; Jia et al., 2016). Figure 6
illustrates the interannual variations of emissions, deposition and $R_{RDN/OXN}$ for China as
well as the more developed USA and Europe (28 countries). The emission data for the
three continents were respectively taken from MEIC, the U.S. Environmental
Protection                                                    Agency                                    (EPA,
https://www.epa.gov/air-emissions-inventories/air-pollutant-emissionstrends-data), and
European Environment Agency (EEA, https://www.eea.europa.eu/themes/air). As
shown in Figure 6a and 6c, the interannual trends in estimated deposition were
basically consistent with those in emissions, with observed reduction for both OXN
and S deposition over the USA and Europe. With the slowdown in economic growth
and the implementation of air pollution control actions for decades (e.g., Clean Air Act
(CAA) in the USA and Convention on Long-range Transboundary Air Pollution
(CLRTAP) in Europe), the emissions of NO$_X$ and SO$_2$ have been reduced by more than
60% and 90% in between 1980 and 2020, respectively (Fowler et al., 2013). However,
as a result of the rapidly growing demand for economic development and energy, the
fossil fuel consumption and fertilizer utilization increased by 3.2 and 2.0 times
during1980-2010 for China, which ultimately led to an increase in the OXN and RDN
deposition from 2005 to 2010 (An et al., 2019; Li, 2020; Liu et al., 2020). Following
developed countries, gradually tightened measures of reducing the acidifying air
pollutants have been launched since 2005, and the deposition began to decline
afterwards.
We selected the periods with fast declines in deposition of OXN and S for the
three continents and compared them in Table 1. The relative changes in deposition
were smaller than those of emissions for all the continents, and greater declines were
found for S for both emissions and deposition than OXN. Compared with Europe and





the USA, China had the smallest benefit of precursor emission abatement on deposition.
For example, the $SO_2$ emissions in the USA, Europe and China had been cut by 78.4%
(2003-2016), 57.6% (2000-2013) and 75.5% (2007-2020) respectively, while S
deposition had declined by 72.5%, 49.9% and 27.0%. This may be caused by a lagging
response of deposition to emission abatement, which is more prominent in China.
Europe and the USA started emission controls earlier than the selected periods, resulted
in a smaller gap between the changes in emissions and deposition afterwards. The
comparison implies that the effect of short-term emission reduction in China would not
immediately be fully reflected in the deposition, but continuous efforts on emission
abatement should be made to achieve substantial reduction in deposition and to further
mitigate ecological risks.
Figure 6d presents the interannual changes of $R_{RDN/OXN}$ for China, USA, and
Europe (28 countries). The $R_{RDN/OXN}$ in China was higher than those in the other two,
with an average of 1.3 in 2005-2020 (0.9 and 1.0 for the USA and Europe during the
same period). As a developing country, China is an important food producing country
in the world, with a long history of agricultural production and planting. Large
agricultural production and relatively weak policy management made China the largest
$NH_3$ emissions in the world, leading to a high proportion of RDN deposition to the
total N deposition (Kang et al., 2016; Liu et al., 2022). In contrast, in developed USA
and Europe with high level of agricultural mechanization and abundant industry and
transportation, the relatively high $NO_X$ emissions compared to $NH_3$ resulted in smaller
$R_{RDN/OXN}$ than China.
Similar temporal changes in $R_{RDN/OXN}$ can be found for USA and China, i.e.,
decline in earlier years and growth afterwards. For USA, the turning point of $R_{RDN/OXN}$
occurred in 1999, 13 years earlier than that of China in 2012. The turning points were
closely associated with the introduction and implementation of $NO_X$ emission controls
for the two countries (CAA Amendments since 1990 for the USA and NAPPCAP since



2013 for China). While RDN in China has been the major species since 2005, the OXN
in the USA was larger than RDN for over 20 years. The $R_{\text{RDN/OXN}}$ kept growing since
2000 and exceeded 1 in 2014, indicating a transition of major N species in the
deposition. Different from China and the USA, $R_{\text{RDN/OXN}}$ in Europe kept declining
since 2000, and being smaller than 1 after 2013. In many European countries with
abundant agricultural activities, the chemical fertilizer and livestock breeding release a
large amount of $NH_3$. Europe attached great importance to the source control of
agricultural pollution, adopted the economic guidance method for agricultural
environmental subsidies, and member states actively assumed the responsibility for
governance for decades (i.e., Common Agriculture Policy, CAP; Zhang et al., 2020).
Therefore, the control of $NH_3$ in Europe was ahead of China, resulting in continuous
reduction in $NH_3$ emissions and thereby $R_{\text{RDN/OXN}}$.

### 514     3.3 Spatial variability in deposition of Nr species and sulfur

Figure 7 shows the spatial distributions of N and S deposition fluxes during
2005-2020. In general, relatively large deposition was found in eastern China with
more population and developed industrial economy (e.g., SE and part of NC in Figure
1). Hotspots of dry deposition were commonly located in the north while wet in the
south. As a joint effect of concentrations and $V_d$, high level of OXN dry deposition was
estimated in areas with high vegetation cover, such as Yunnan and Fujian province. For
S dry deposition, coal-fired boilers for power and heating were intensively distributed
in the north, leading to abundant $SO_2$ emissions and thereby dry deposition.
Furthermore, the relatively stable weather conditions with less convection in the north
was unfavorable to the dispersion and dilution of pollutants. The emissions were thus
liable to be deposited locally. For RDN, the agricultural production, animal husbandry
and biomass burning in NC and the northern part of SE led to relatively $NH_3$ emissions
and thereby high dry deposition. The more acidic and humid soils in the south made
$NH_3$ more difficult to release, resulting in lower dry deposition compared to the north.



Large wet deposition was mainly found in the south of China associated with the
uneven distribution of precipitation. In summer, the air masses in the western Pacific
Ocean and the South China Sea were affected by the southeast and southwest monsoon,
significantly increasing the rainfall in southeast China. For the total deposition (wet
plus dry), the high deposition of OXN and S were located in SE, while RDN and total
N were mainly concentrated in NC and the north of SE.

As shown in Table S5 in the Supplement, the $R_{dry/wet}$ of N and S in the eastern

China (SE+NC with Inner Mongolia excluded) was smaller than that in western China
(NW+TP), attributed mainly to the large precipitation in the former. Given the dry
climate and less anthropogenic activities, the pollution was mainly transported by
atmospheric turbulence and removed from the atmosphere by dry deposition in western
country. The $R_{dry/wet}$ of TP was the highest out of the six regions, with 2.6 and 3.7 for
total N and S, respectively. The $R_{dry/wet}$ in NE, NW and NC was generally higher than
that in the south (SE and SW), resulting also from the abundant precipitation in the
south. Higher $R_{RDN/OXN}$ was found in the west (e.g., NW and TP) and lower in the east
(Table S5), as more developed industry in the east resulted in relatively large $NO_X$
emissions and thereby OXN deposition, while farming and animal husbandry
dominated the economy in the west, leading to substantial $NH_3$ emissions.

Figure 8 and Table 2 compare the relative changes of total deposition (wet plus

dry) of different species for eastern, western and whole country. The interannual
changes of deposition for all species were smaller than that of emissions (Table 2),
reconfirming lagging response of deposition to changing emissions as mentioned in
Section 3.2. During the period when emissions declined rapidly, the change of
deposition has not yet occurred. The relative changes for N and S deposition in eastern
China were generally larger than the whole country, indicating the effectiveness of
extremely stringent emission controls on those regions with abundant emissions from
industrial and traffic sources. The OXN deposition for all the concerned regions shows





an invert "V" pattern over time, consistent with the progress of $NO_X$ emissions control
(Figure 8a). The relative annual changes in eastern China (9% in 2005-2012 and -12%
in 2012-2020) were generally greater than in western (4% in 2005-2012 and -5% in
2012-2020). More specifically, the turning point for western China was later than the
East, likely resulting from later implementation of emission control policies. Most
measures were first implemented in the highly developed key regions in east and then
applied more widely afterwards. As shown in the Figure 8b and Table 2, RDN
deposition was relatively stable before 2012, and the temporal changes in eastern and
western China were generally consistent with each other. The lack of comparable
control policies for $NH_3$ and strict policy of acid precursors likely explained the
increasing trend in RDN afterwards, with 9% in eastern and 10% in western China
between 2012 and 2020. The biggest reduction was achieved for S deposition, and the
decline in eastern China was faster than that in the western (Figure 8c). Attributable to
the earlier and broader use of FGD at coal combustion sources, greater abatement of
$SO_2$ emissions was achieved than $NO_X$ or $NH_3$ over the past decade, leading to the
faster reduction in S deposition than in OXN or RDN (Table 2). In addition, the
reduction during 2012-2020 (28%, 18% and 21% for the eastern, western and the
whole country, respectively) was clearly larger than that during 2005-2012 (3%, 9%
and 7%, respectively), indicating the greatly improved $SO_2$ controls compared to
earlier years.

The ratio of deposition to emissions (D/E) is used to analyze the interactions

between the pollutant sources and sinks. Figure 9a shows the annual mean D/E ratios
during 2005-2020 by species and region. The D/E in eastern China (e.g., NC and SE)
was generally smaller than in western China (NW, SW and TP). The low D/E identified
those regions as the major sources of air pollutants due mainly to their intensive
emissions, likely influencing air pollution levels in surrounding regions. With less
industry, energy consumption and population, by contrast, western China received



relatively high deposition compared to local emissions, resulting in large D/E. The very
high ratio of D/E indicated that TP was strongly influenced by regional pollution
transport. The D/Es of RDN in the six regions were higher than that of OXN and sulfur
(except for TP). Due to its relatively short life time, most of $NH_3$ deposits near the
source area, while stronger transport and chemical reaction may occur for $NO_X$ and
$SO_2$ given their longer life time. Significantly positive correlations were found between
regional deposition and emissions for all the concern species, with $R^2$ estimated at 0.81,
0.92, and 0.78 for OXN (Figure 9b), RDN (Figure 9c), and S (Figure 9d), respectively.
The result implies that the N and S deposition to the six regions were strongly
dependent on the spatial pattern of anthropogenic emissions.
The annual emissions, deposition and D/E by land use type were displayed in
Table S6 in the supplement. High deposition was commonly found in areas with high
energy consumption and large emissions, such as urban and construction sites.
Associated with different human activities, moreover, the D/E for sulfur and OXN
were smaller in urban regions than those in rural ones, whereas that for RDN was
slightly larger in urban areas. Transportation and industries resulted in larger $NO_X$ and
$SO_2$ emissions in urban locales and agricultural activities enhanced $NH_3$ in rural ones.
Figure 10 shows the spatial distribution of multi-year average deposition by
season, which was influenced jointly by varying meteorology and emissions. Basically,
larger deposition was found in summer than that in winter, and the seasonal difference
was particular bigger for N. The deposition in summer was estimated to be 1.9 and 1.6
times in winter for OXN and RDN, respectively, while the ratio was much smaller at
1.1 for S. The hotspot of deposition was commonly found in NC and northern SE in
summer, while it moved to central SE in winter attributed partly to the prevailing
northwesterly wind.



### 3.4 Uncertainties


609   Uncertainties existed in current analysis. First, the estimated dry deposition or $V_d$
could not be fully examined with sufficient data from direct observation, attributed
mainly to the lack of field measurements. Micrometeorological methods can be used
for direct observation of dry deposition, including eddy correlation method, gradient
method and relaxation vortex accumulation method. Due to the need for extremely fast
response instruments and uniform underlying surfaces, those methods have not yet
been widely applied in a long-term and extensive manner. Second, error may come
from ground-level monitoring data. We collected available data from different
monitoring networks, and ignored the difference in observed deposition from diverse
methods of sample collection and measurement. Moreover, current RF model relied on
the data from observation sites, most of which are located in the eastern China with
dense population and developed economy. The model accuracy for remote areas (such
as NW and TP) should be further evaluated when more observation data get available
for those areas. Third, there was additional uncertainty in the estimation of sulfate dry
deposition, as there were limited observed ambient concentrations of sulfate available
for estimation of dry deposition, and CTM had to be applied. Furthermore, bulk
deposition obtained from the open precipitation gauge contains part of dry deposition
and therefore likely overestimate actual wet deposition. The bias varied by region and
was hard to be quantified at the national level. For example, research indicated that the
dry deposition accounted for around 20% of the bulk deposition based on observation
at three rural stations on the North China Plain, and this contribution could reach 39%
in urban areas (Zhang et al., 2015; Zhang et al., 2008). Along with continuous
development of monitoring networks and increasing availability of deposition data for
diverse species, those uncertainties can be further reduced and more accurate
deposition estimation can be expected.



**4. Conclusions**
We developed a full N and S deposition dataset for mainland China at the
horizontal resolution of 0.25° for 2005-2020, combining the ground-level observations,
satellite-derived VCDs, meteorological and geographic information, and CTM. Based
on the newly developed RF method, the annual average dry deposition of OXN, RDN
and S in China was estimated at 10.4, 14.4 and 16.7 kg N/S ha$^{-1}$ yr$^{-1}$, while the total
deposition reached 15.2, 20.2 and 25.9 kg N/S ha$^{-1}$ yr$^{-1}$, respectively, with the wet
deposition estimated with a GAM model included. The $R_{\text{dry/wet}}$ of N kept relatively
stable at the beginning and then gradually increased, especially for RDN, while that of
S declined for over 10 years and then slightly increased. Within the whole study period,
$R_{\text{RDN/OXN}}$ was estimated to be greater than 1 and clearly larger than that of the USA and
Europe, with a continuous decline from 2005 to 2011 and a growth afterwards. The
frequent agricultural activities and relatively weak management of manure have
resulted in abundant NH$_3$ emissions and thereby a high proportion of RDN deposition.
Improved NO$_X$ emission control was the main reason for the elevated $R_{\text{RDN/OXN}}$ for
recent years. Compared with Europe and the USA, China had the smallest benefit of
precursor emission reduction on deposition. The prominent lagging response of
deposition to emission abatement requires a continuous long-term emission control
efforts to substantially reduce atmospheric deposition. As a joint effect of emissions
and individual meteorological factors, a downward gradient from east to west was
found for dry deposition of OXN while from north to south for S. The wet deposition
frequently occurred in the south of China, associated with the spatial distribution of
rainfall. The deposition of OXN and S declined faster in eastern China than that in the
west after 2012, indicating the effectiveness of extremely strict emission control in
developed areas with abundant emissions from industry and transportation. The D/E in
eastern China was generally smaller than that in west, as the former was the major
sources of air pollutants and the latter received relatively high deposition through
regional transport. At the national scale, the deposition strongly depended on the



spatial pattern of anthropogenic emissions within the regions. The current study
broadens the scientific understanding of China's long-term changes in deposition of
typical atmospheric species, as well as the influences of human activities and emission
controls. More observation and modeling work is recommended for in-depth analyses
on the complicated and changing relationship between emissions and deposition for
specific species, as well as the consequent varying effects on ecosystem.
**Data availability**
The multiyear deposition data by species at the horizontal resolution of 0.25º will be
available at http://www.airqualitynju.com/En/Data/List/Datadownload once the paper
is published.
**Author contributions**
KZhou developed the methodology, conducted the research, performed the analyses
and wrote the draft. YZhao developed the strategy, designed the research and revised
the manuscript. LZhang and MMa provided the support of air quality modeling. WXu
and XLiu provided the support of NNDMN data.
**Competing interests**
The authors declare that they have no conflict of interest.
**Acknowledgements**
This work was sponsored by the Natural Science Foundation of China (42177080) and
the Key Research and Development Programme of Jiangsu Province (BE2022838). We
acknowledge Qiang Zhang from Tsinghua University for the emission data (MEIC),
and Jintai Lin from Peking University for the satellite data (POMINO v2).

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



**Figure captions**
Figure 1 The research domain of this study. The pink points represent CNEMC and the
green points represent NNDMN. The Qinling-Huaihe Line is the boundary between the
north and the south of the country.
Figure 2 Methodology framework to estimate dry and wet deposition of this study. The
blue process shows the four steps to establish the RF model. The orange process shows
the three steps in establishing a GAM model. See Sections 2.2 to 2.3 of the method
section in the text for the acquisition of the preliminary data set.
Figure 3 Comparison of deposition between this study and other literatures for dry (a),
wet (b) and total deposition (c). The black cross and the pentagram are the average of
literature-reported results and the multi-year average of this study, respectively. The
boxplots represent the dispersion of deposition collected from literatures. The central
horizontal line, the upper side line, and the lower side line of the box represent the
median value, the upper quartile (75th Quantile, Q3) and the lower quartile (25th
Quantile, Q1). The vertical line extending out of the box represents 1.5 times the
interquartile interval (IQR, i.e., Q3-Q1), and the horizontal lines represent the upper
limit (Q3+1.5IQR) and the lower limit (Q1-1.5 IQR).
Figure 4 Contribution of different forms and species to the estimated total N and S
deposition in China.
Figure 5 The interannual variability of N and S deposition, emissions and component
proportion in China from 2005 to 2020. The emission data over China were taken from
MEIC.
Figure 6 The interannual variations of emissions, deposition and RDN/OXN for China,
28 Europe countries (EU) and the USA. All the data are relative to the 2005 levels. The
grey dotted lines are a visual guidance for 1.0 on each of the y axes. (a) $NO_X$ emissions
and OXN deposition; (b) $NH_3$ emissions and RDN deposition; (c) $SO_2$ emissions and
sulfur deposition; and (d) RDN/OXN.
Figure 7 The spatial distributions of N and S deposition flux in 2005-2020.
Figure 8 The interannual variations and relative changes of deposition of OXN (a),
RDN (b) and sulfur (c) by region. All the data are relative to the 2005 levels. The



orange line represents eastern China (SE+NC with Inner Mongolia excluded, see
Figure 1 for the region definitions), the blue line represents western China (NW+TP),
and the red line represents the average level of whole China.
Figure 9 Annual mean D/E ratio of OXN, RDN and sulfur from 2005 to 2020 in
different regions (a) and linear relationship between regional deposition and emissions
(b-d).
Figure 10 The spatial distribution of multi-year seasonal variation of the total
deposition across 2005-2020.



**Tables**
**Table 1 Comparison of relative change rates of emissions and deposition in the**
**process of pollution control in China, Europe and the USA.** The starting and ending
time was selected according to the period of the fastest decline of deposition in China,
and the time period of emission decline was selected according to the reference
deposition. The emission data were respectively taken from MEIC, the European
Environment Agency (EEA, https://www.eea.europa.eu/themes/air), and U.S.
Environmental Protection Agency (EPA,
https://www.epa.gov/air-emissions-inventories/air-pollutant-emissionstrends-data,
while deposition data from European Monitoring and Evaluation Programme (EMEP,
https://projects.nilu.no/ccc/index.html) for Europe and Clean Air Status and Trends
Network (CASTNET, https://www.epa.gov/castnet) and National Atmospheric
Deposition Program (NADP,
https://nadp.slh.wisc.edu/networks/national-trends-network/) for the USA.

| Relative change | Emissions | | | |
| --- | --- | --- | --- | --- |
| | NO$_X$ | | SO$_2$ | |
| The USA | -35.9% | (2003-2011) | -78.4% | (2003-2016) |
| Europe | -17.3% | (2000-2008) | -57.6% | (2000-2013) |
| China | -32.2% | (2012-2020) | -75.5% | (2007-2020) |
| | Deposition | | | |
| | OXN | | S | |
| The USA | -26.0% | (2003-2011) | -72.5% | (2003-2016) |
| Europe | -11.1% | (2000-2008) | -49.9% | (2000-2013) |
| China | -7.1% | (2012-2020) | -27.0% | (2007-2020) |






**Table 2 The interannual changes in deposition and emissions of N and S by**
**regions for 2005–2020.** Eastern China includes NC (Inner Mongolia excluded) and SE,
and western China includes TP and NW (see Figure 1 for the region definitions). P1
and P2 indicate 2005–2012 and 2012–2020, respectively.

| Interannual change (units: kg N/S ha$^{-1}$ yr$^{-1}$) | | Whole China | | Eastern China | | Western China | |
|---|---|---|---|---|---|---|---|
| | | P1 | P2 | P1 | P2 | P1 | P2 |
| Emissions | NO$_X$ | 0.60 | -0.42 | 1.12 | -1.33 | 0.63 | -0.24 |
| | NH$_3$ | 0.08 | -0.21 | 0.08 | -0.83 | 0.09 | -0.02 |
| | SO$_2$ | -0.39 | -1.24 | -2.98 | -4.62 | 0.01 | -0.89 |
| Deposition | Total OXN | 0.09 | -0.15 | 0.22 | -0.41 | 0.07 | -0.08 |
| | Total RDN | 0.05 | 0.06 | 0.06 | 0.28 | 0.05 | 0.22 |
| | Total N | 0.14 | -0.09 | 0.28 | -0.14 | 0.13 | 0.14 |
| | Total S | -0.29 | -0.82 | -0.34 | -1.55 | -0.29 | -0.60 |
| Relative annual change to 2005 (P1) or 2012 (P2) | | P1 | P2 | P1 | P2 | P1 | P2 |
| Emissions | NO$_X$ | 49% | -31% | 17% | -25% | 110% | -29% |
| | NH$_3$ | 7% | -15% | 2% | -22% | 17% | -3% |
| | SO$_2$ | -13% | -72% | -25% | -73% | 10% | -74% |
| Deposition | Total OXN | 5% | -7% | 9% | -12% | 4% | -5% |
| | Total RDN | 3% | 3% | 5% | 9% | 3% | 10% |
| | Total N | 4% | -2% | 7% | -2% | 3% | 1% |
| | Total S | -7% | -21% | -3% | -28% | -9% | -18% |




**Figures**
**Figure 1**

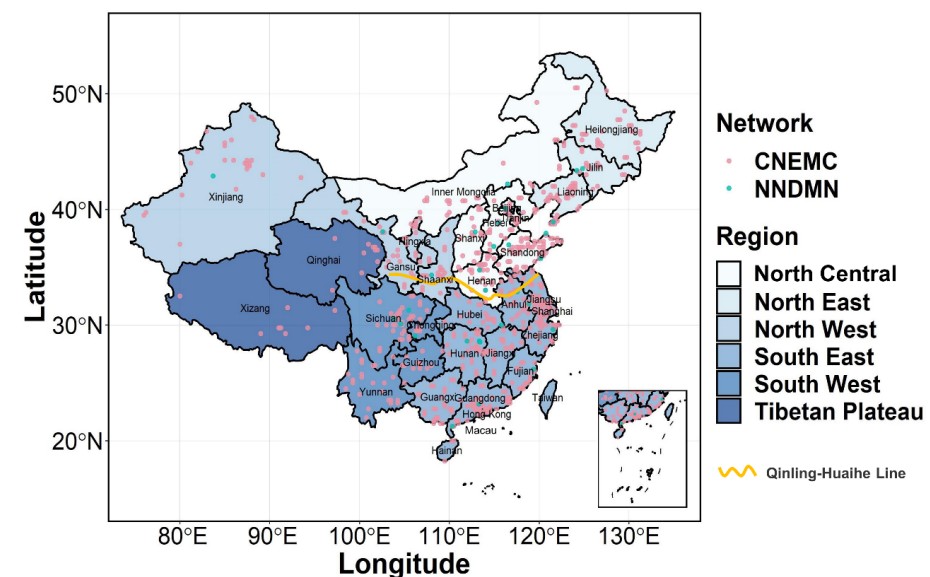





**Figure 2**

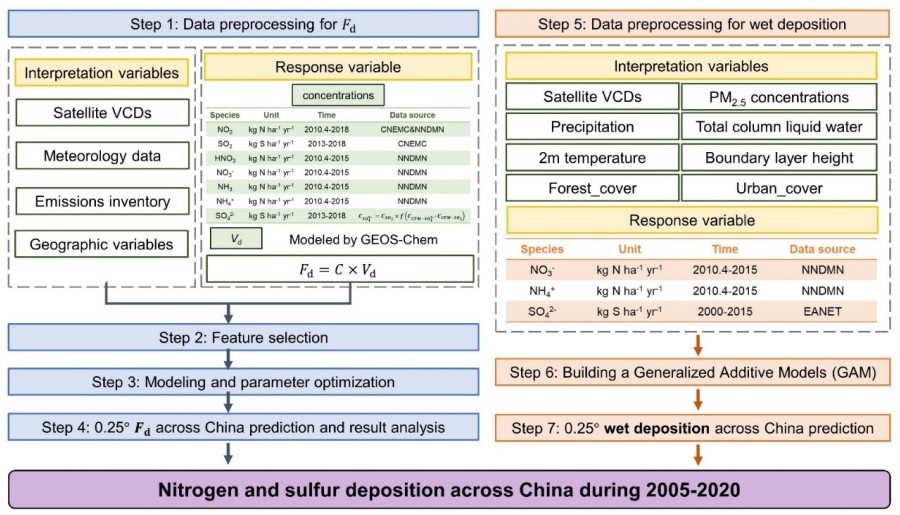




**Figure 3**

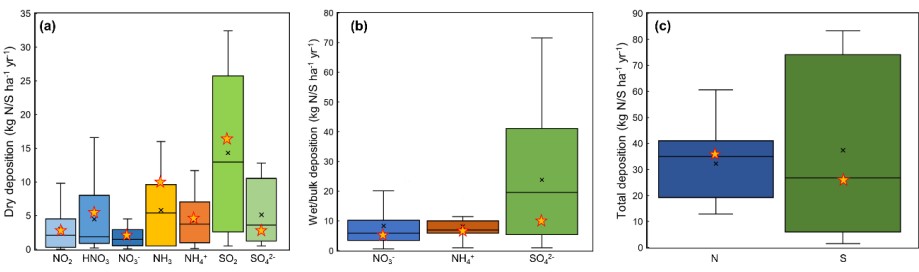







**Figure 4**

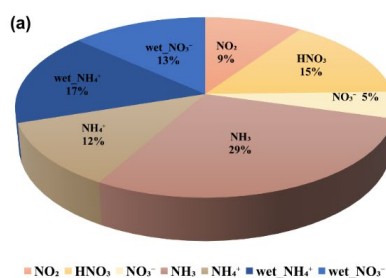
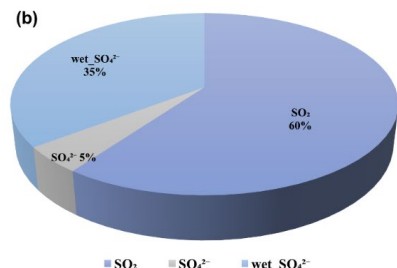




**Figure 5**

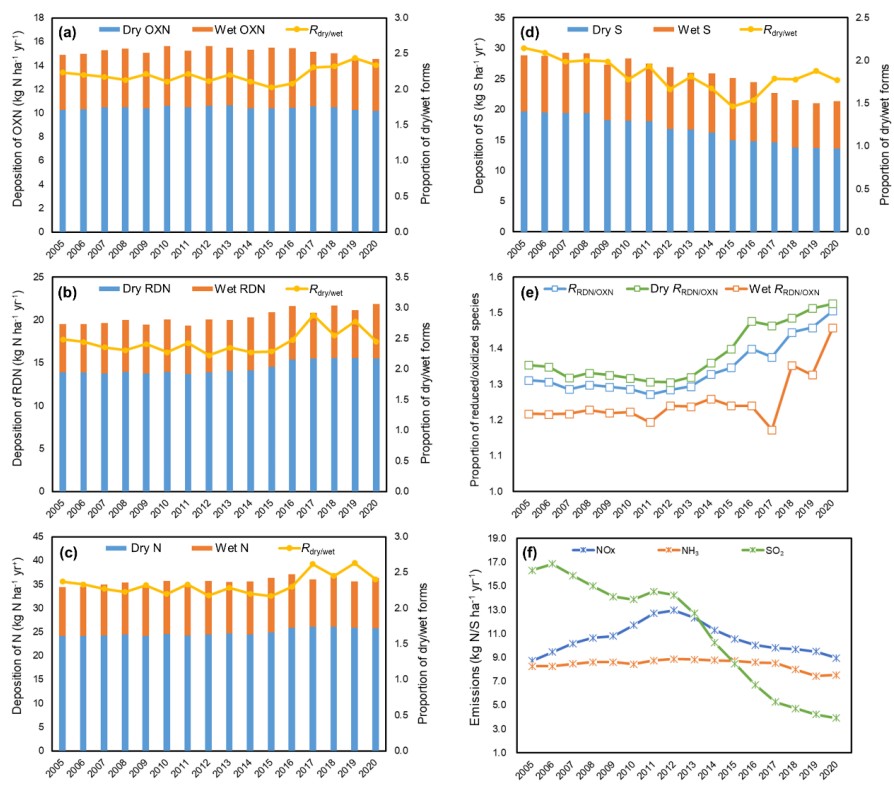





**Figure 6**

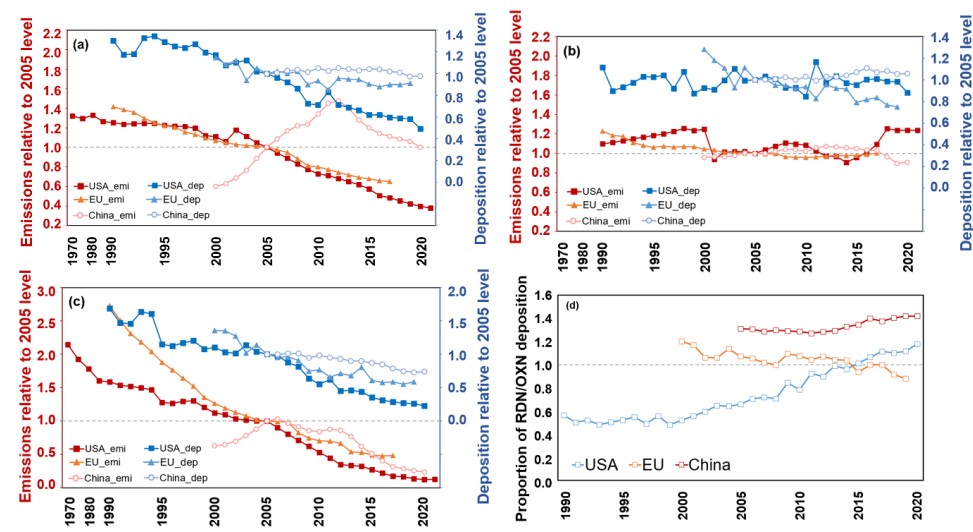





**Figure 7**

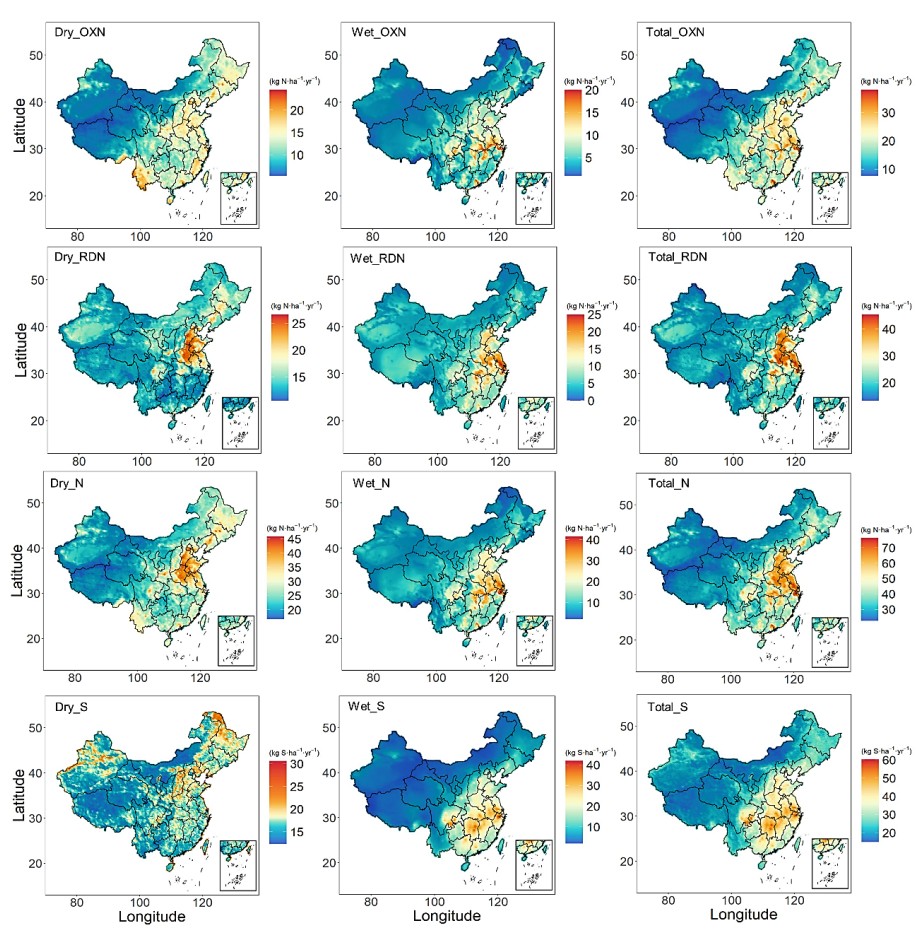







**Figure 8**

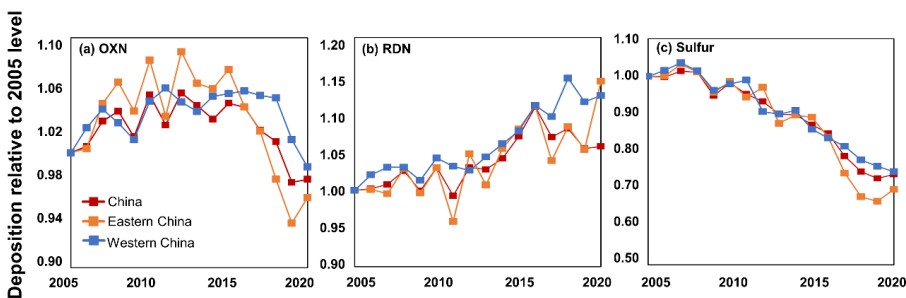





**Figure 9**

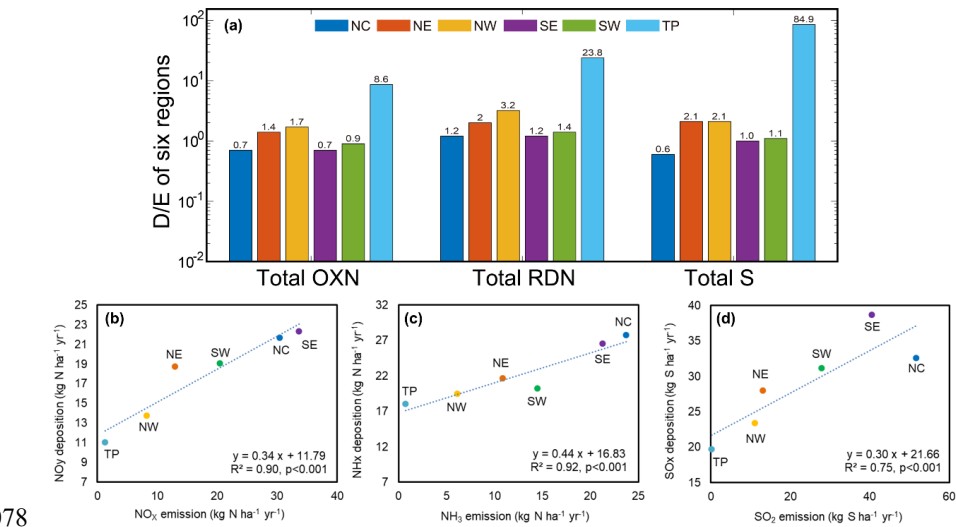





**Figure 10**

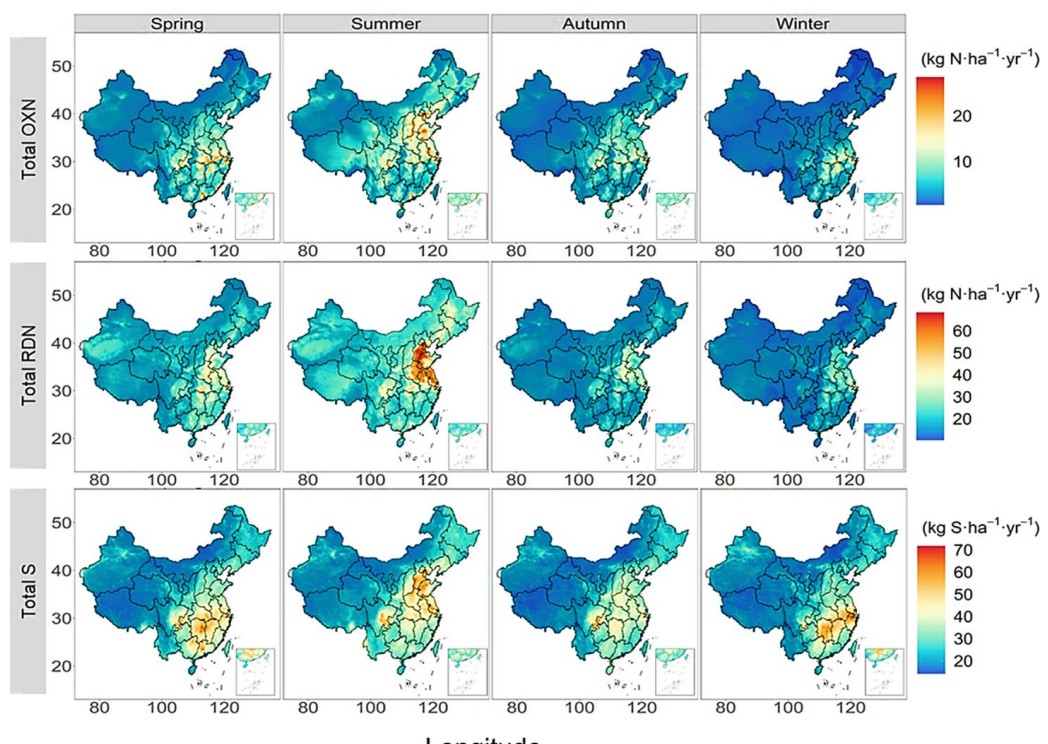

