# Peer review of "Estimating nitrogen and sulfur deposition across China during 2005-2020 based on multiple statistical models"

_EGUsphere, 2023_

## Author Comment (AC1)

**SUPPLEMENT FOR THE RESPONSE TO REVIEWER #1**

**Number of tables:4 Number of figures: 3**

**Table list**

Table R1 in the Main revisions and response: Comparisons of total N deposition fluxes between our study and Vet et al (2014) (kg N/S  $ha^{-1}$  yr-1).

Table S3 in the revised supplement: The modeled  $V_d$  for different land use categories (cm s-1).

Table S5 in the revised supplement: The Mann-Kendall test for the trend of  $R_{dry/wet}$  of N and S in 2005-2020. The z-value represents the standard normal statistic, and the p-value represents the generalization. The former indicates the trend, while the latter indicates statistical significance. P1 and P2 indicate 2005–2015 and 2015–2020, respectively.

Table S6 in the revised supplement (Table S4 in the original submission): Comparisons of total deposition fluxes of different species between our study in China and two networks in other countries (kg N/S ha-1 yr-1).

**Figure list**

Figure 2 in the revised manuscript: Methodology framework to estimate dry and wet deposition of this study. The blue process shows the four steps to establish the RF model. The orange process shows the three steps in establishing a GAM model. See Sections 2.2 to 2.3 of the method section in the text for the acquisition of the preliminary data set.

Figure S4 in the revised supplement (Figure S3 in the original submission): China average total precipitation from ECMWF: https://apps.ecmwf.int/datasets/data/interim-full-daily/levtype=sfc/.

Figure S5 in the revised supplement: in the revised supplement: The monthly means of the modeled dry deposition velocity of N and S during 2013-2020.

 Table R1 in the Main revisions and response: Comparisons of total N deposition fluxes between our study and Vet et al (2014) (kg N/S ha-1 yr-1).

| Reference        | Ν                                                                             | S                                                                                     |
|------------------|-------------------------------------------------------------------------------|---------------------------------------------------------------------------------------|
| This study       | 16.0                                                                          | 16.1                                                                                  |
| Vet et al (2014) | 0.4-20.0                                                                      | 4.0-23.4                                                                              |
| This study       | 6.1                                                                           | 4.1                                                                                   |
| Vet et al (2014) | 2.0-28.1                                                                      | 4.0-32.0                                                                              |
|                  | Reference
This study
Vet et al (2014)
This study
Vet et al (2014) | ReferenceNThis study16.0Vet et al (2014)0.4-20.0This study6.1Vet et al (2014)2.0-28.1 |

| Land use categories       | HNO 3 | NH3  | NH 4 | NO 2 | NO 3 | SO 2 | SO 4 |
|---------------------------|------------------|------|-----------------|-----------------|-----------------|-----------------|-----------------|
| Paddy fields              | 1.63             | 0.47 | 0.14            | 0.20            | 0.14            | 0.45            | 0.14            |
| Dry land                  | 1.42             | 0.42 | 0.16            | 0.17            | 0.16            | 0.41            | 0.16            |
| Forestland                | 2.55             | 0.49 | 0.14            | 0.24            | 0.14            | 0.46            | 0.14            |
| Shrub forest              | 1.83             | 0.45 | 0.16            | 0.21            | 0.16            | 0.43            | 0.16            |
| Sparse forestland         | 1.96             | 0.47 | 0.15            | 0.22            | 0.15            | 0.44            | 0.15            |
| Other forestland          | 2.17             | 0.53 | 0.14            | 0.22            | 0.14            | 0.52            | 0.14            |
| High coverage grassland   | 1.29             | 0.36 | 0.18            | 0.12            | 0.18            | 0.36            | 0.18            |
| Medium coverage grassland | 1.05             | 0.34 | 0.18            | 0.09            | 0.18            | 0.34            | 0.18            |
| Low coverage grassland    | 0.88             | 0.31 | 0.17            | 0.04            | 0.17            | 0.31            | 0.17            |
| River channel             | 1.17             | 0.38 | 0.15            | 0.13            | 0.15            | 0.37            | 0.15            |
| Lakes                     | 0.93             | 0.32 | 0.19            | 0.07            | 0.19            | 0.32            | 0.19            |
| Reservoir pond            | 1.37             | 0.43 | 0.14            | 0.14            | 0.14            | 0.43            | 0.14            |
| Permanent glacial snow    | 0.59             | 0.27 | 0.16            | 0.03            | 0.16            | 0.28            | 0.16            |
| Tidal-flat                | 1.06             | 1.01 | 0.07            | 0.02            | 0.07            | 1.02            | 0.07            |
| Beach land                | 1.30             | 0.37 | 0.16            | 0.12            | 0.16            | 0.36            | 0.16            |
| Urban land use            | 1.37             | 0.44 | 0.15            | 0.14            | 0.15            | 0.44            | 0.15            |
| Rural settlements         | 1.27             | 0.40 | 0.15            | 0.14            | 0.15            | 0.40            | 0.15            |
| Other construction land   | 1.40             | 0.47 | 0.14            | 0.16            | 0.14            | 0.47            | 0.14            |
| Sand                      | 0.87             | 0.30 | 0.10            | 0.02            | 0.10            | 0.30            | 0.10            |
| Gobi                      | 0.98             | 0.30 | 0.10            | 0.02            | 0.10            | 0.30            | 0.10            |
| Saline alkali soil        | 0.87             | 0.31 | 0.15            | 0.04            | 0.15            | 0.31            | 0.15            |
| Swamp land                | 1.61             | 0.38 | 0.15            | 0.14            | 0.15            | 0.37            | 0.15            |
| Bare land                 | 1.10             | 0.30 | 0.10            | 0.03            | 0.10            | 0.30            | 0.10            |
| Bare rock                 | 0.88             | 0.30 | 0.14            | 0.03            | 0.14            | 0.30            | 0.14            |
| Other unused land         | 0.72             | 0.29 | 0.18            | 0.04            | 0.18            | 0.30            | 0.18            |

Table S3 in the revised supplement: The modeled  $V_d$  for different land use categories (cm s-1).

Note: Land-Use and Land-Cover Change (LUCC) data were obtained from Resource and Environment Data Cloud Platform (http://www.resdc.cn/), generated by manual visual interpretation of Landsat TM/ETM remote sensing images.

Table S5 in the revised supplement: The Mann-Kendall test for the trend of  $R_{dry/wet}$  of N and S in 2005-2020. The z-value represents the standard normal statistic, and the p-value represents the generalization. The former indicates the trend, while the latter indicates statistical significance. P1 and P2 indicate 2005–2015 and 2015–2020, respectively.

| Species | ٥X     | KN    | RI     | DN    | Ν      | 1     | S      | 5     |
|---------|--------|-------|--------|-------|--------|-------|--------|-------|
| Period  | P1     | P2    | P1     | P2    | P1     | P2    | P1     | P2    |
| Z       | -2.024 | 2.254 | -2.024 | 1.879 | -2.336 | 1.879 | -3.270 | 1.127 |
| р       | 0.043  | 0.024 | 0.043  | 0.060 | 0.020  | 0.060 | 0.001  | 0.260 |

Note: Negative and positive z-value indicate a downward and upward trend in the time series, respectively;

p<0.01 indicates a significance level of 99%.

Table S6 in the revised supplement (Table S4 in the original submission): Comparisons of total deposition fluxes of different species between our study in China and two networks in other countries (kg N/S ha-1 yr-1).

|        | Period    | RDN  | OXN  | Ν    | S    |
|--------|-----------|------|------|------|------|
| USA    | 1990-2020 | 2.7  | 13.3 | 16.0 | 16.1 |
| Europe | 2000-2019 | 1.4  | 4.7  | 6.1  | 4.1  |
| China  | 2005-2020 | 20.2 | 15.2 | 35.4 | 25.9 |

Figure 2 in the revised manuscript: Methodology framework to estimate dry and wet deposition of this study. The blue process shows the four steps to establish the RF model. The orange process shows the three steps in establishing a GAM model. See Sections 2.2 to 2.3 of the method section in the text for the acquisition of the preliminary data set.

---

## Author Comment (AC2)

**SUPPLEMENT FOR THE RESPONSE TO REVIEWER #2**

**Number of tables:1 Number of figures: 3**

**Table list**

Table S4 in the revised supplement (Table S3 in the original submission): Comparison of the annual  $F_d$  of N and S in this and other studies (kg N/S ha-1 yr-1).

**Figure list**

Figure 5 in the revised manuscript: The interannual variability of N and S deposition, emissions and component proportion in China from 2005 to 2020. The emission data over China were taken from MEIC.

Figure 7 in the revised manuscript: The spatial distributions of N and S deposition flux in 2005-2020.

Figure 9 in the revised manuscript: Annual mean D/E ratio of OXN, RDN and sulfur from 2005 to 2020 in different regions (a) and linear relationship between regional deposition and emissions (b-d).

Table S4 in the revised supplement (Table S3 in the original submission): Comparison of the annual  $F_d$  of N and S in this and other studies (kg N/S ha-1 yr-1).

| Reference              | Study region    | Research scale   | Study     |         | Dry deposition   |          |                 |                       |        |                   | Wet/bulk deposition   |          |                   | Total deposition |      |
|------------------------|-----------------|------------------|-----------|---------|------------------|----------|-----------------|-----------------------|--------|-------------------|-----------------------|----------|-------------------|------------------|------|
|                        |                 |                  | period    | $NO_2$  | HNO 3 | $NO_3^-$ | NH 3 | $\mathrm{NH_{4}^{+}}$ | $SO_2$ | SO4 2- | $\mathrm{NH_{4}^{+}}$ | $NO_3^-$ | SO4 2- | Ν                | S    |
| This study             | China           | Grid level       | 2005-2020 | 3.4     | 5.3              | 1.7      | 10.3            | 4.2                   | 15.5   | 1.2               | 3.3                   | 4.6      | 6.4               | 32.9             | 23.1 |
| Nowlan et al. (2014)   | China           | Grid level       | 2005-2007 | 0.2     |                  |          |                 |                       |        |                   |                       |          |                   |                  |      |
| Lye and Tian (2007)    | China           | Grid level       | 2003      | 2.9     |                  |          |                 |                       |        |                   | 7.1                   | 2.8      |                   | 12.9             |      |
| Jia et al. (2014)      | China           | Grid level       | 1980-2010 |         |                  |          |                 |                       |        |                   |                       |          |                   |                  |      |
| Jia et al. (2016)      | China           | Grid level       | 2005-2014 | 0.6     | 1.1              | 0.1      | 5.4             | 0.3                   |        |                   | 13.9                  |          |                   |                  |      |
| Tan et al. (2022)      | China           | Grid level       | 2010      |         |                  |          |                 |                       |        |                   | 4.2                   | 3.4      |                   |                  |      |
| Itahashi et al. (2018) | China           | Grid level       | 2010      |         |                  |          |                 |                       |        |                   | 3.5                   | 2.5      |                   |                  |      |
| Zhao et al. (2017)     | China           | Grid level       | 2008-2012 | 0.3     | 1.7              | 1.0      | 0.5             | 2.6                   |        |                   | 6.6                   | 3.4      |                   | 18.1             |      |
| Xu et al. (2015)       | China           | 43 sites         | 2010-2014 | 0.2-9.8 | 0.2-16.6         | 0.1-4.5  | 0.5-16.0        | 0.1-11.7              |        |                   | 1.0-19.1              | 0.5-20.1 |                   | 39.9             |      |
| Xu et al. (2019)       | China           | 32 sites         | 2010-2015 | 3.1     | 5.2              | 1.4      | 9.6             | 3.7                   |        |                   | 11.4                  | 10.3     |                   |                  |      |
| Wen et al. (2020)      | China           | 66 sites         | 2011-2018 |         |                  | 22.5     |                 |                       |        |                   | 19.4                  |          |                   |                  |      |
| Pan et al. (2012)      | North China     | 10 sites         | 2007-2010 | 0.8-4.5 |                  | 2.2-3.1  | 8.1-64.2        | 1.7-5.5               |        |                   | 10.3-22.0             | 3.4-10.2 |                   | 60.6             |      |
| Pan et al. (2013)      | North China     | 10 sites         | 2007-2010 |         |                  |          |                 |                       | 32.4   | 12.8              |                       |          | 19.6              |                  | 64.8 |
| Zhu et al. (2015)      | China           | Grid level       | 2013      |         |                  |          |                 |                       |        |                   | 7.3                   | 5.9      |                   |                  |      |
| Yu et al. (2016)       | China           | 43 sites         | 2009-2014 |         |                  |          |                 |                       |        |                   |                       | 32.9     | 116.0             |                  |      |
| Yu et al. (2019)       | China           | Grid level       | 1985-2015 | 0.8     | 2.0              | 2.7      | 0.5             | 4.3                   |        |                   | 5.9                   | 4.2      |                   | 20.4             |      |
| Li et al. (2019)       | China           | Grid level       | 2010      |         |                  |          |                 |                       |        |                   |                       |          | 71.5              |                  |      |
| Li et al. (2020)       | China           | Grid level       | 2011-2016 |         |                  |          |                 |                       |        |                   | 5.9                   | 13.3     | 33.4              |                  |      |
| Liu et al. (2016a)     | Southwest China | 1 site           | 2003-2013 |         |                  |          |                 |                       |        |                   | 17.5                  | 8.2      | 21.7              |                  |      |
| Liu et al. (2016b)     | China           | 225 data records | 2003-2014 |         |                  |          |                 |                       |        |                   | 6.8                   | 5.4      |                   |                  |      |
| Liu et al. (2016c)     | China           | 174 sites        | 2000-2013 |         |                  |          |                 |                       |        |                   |                       |          | 23.0              |                  |      |
| Liu et al. (2017a)     | China           | Grid level       | 2010-2012 |         |                  |          |                 |                       |        |                   |                       | 5.8      |                   |                  |      |

| Reference                 | Study region    | Research scale | Study     | Dry deposition  |                  |          |                 |                       |                 |                   | Wet/bulk deposition   |          |                   | Total deposition |          |
|---------------------------|-----------------|----------------|-----------|-----------------|------------------|----------|-----------------|-----------------------|-----------------|-------------------|-----------------------|----------|-------------------|------------------|----------|
|                           |                 |                | period    | NO 2 | HNO 3 | $NO_3^-$ | $\mathrm{NH}_3$ | $\mathrm{NH_{4}^{+}}$ | SO 2 | SO4 2- | $\mathrm{NH_{4}^{+}}$ | $NO_3^-$ | SO4 2- | Ν                | S        |
| Liu et al. (2017b)        | China           | Grid level     | 2012      |                 |                  | 1.5      |                 |                       |                 |                   |                       |          |                   |                  |          |
| Liu et al. (2021)         | China           | Grid level     | 2008-2016 |                 |                  |          |                 |                       |                 |                   | 6.5                   |          |                   |                  |          |
| Luo et al. (2016)         | China           | 16 sites       | 2010-2012 |                 |                  |          |                 |                       | 2.3-26.5        | 0.5-3.4           |                       |          |                   |                  |          |
| Ge et al. (2014)          | China           | Grid level     | 2007      |                 |                  |          |                 |                       |                 |                   | 9.1                   | 9.1      | 48.8              | 35.0             | 83.3     |
| Kuribayashi et al. (2012) | China           | 6 sites        | 2001-2005 |                 |                  |          |                 |                       | 23.5            | 3.8               |                       |          |                   | 49.4             |          |
| Zhang et al. (2017)       | China           | Grid level     | 2007-2014 | 0.005-8.54      |                  |          |                 |                       |                 |                   |                       |          |                   |                  |          |
| Zhou et al. (2021)        | China           | Grid level     | 2013-2018 | 2.1-3.1         |                  |          |                 |                       | 7.5-18.4        |                   |                       |          |                   |                  |          |
| Qiao et al. (2015a)       | Sichuan, China  | 1 site         | 2010-2011 |                 |                  |          |                 |                       |                 |                   | 1.4                   | 1.3      | 8.1               |                  |          |
| Qiao et al. (2015b)       | Sichuan, China  | Grid level     | 2010-2011 |                 |                  |          |                 |                       |                 |                   |                       | 0.3      | 2.8               |                  |          |
| Zhang et al. (2022)       | Tibetan Plateau | 27 sites       |           |                 |                  |          |                 |                       |                 |                   |                       |          |                   |                  |          |
| Larssen et al. (2011)     | South China     | 4 sites        | 2001-2004 |                 |                  |          |                 |                       |                 |                   | 0.4-0.9               | 0.2-0.5  | 0.9-1.9           | 0.4-2.5          | 1.5-10.5 |
| Jiang et al. (2020)       | Hunan, China    | 5 sites        | 2015-2016 |                 |                  |          |                 |                       | 8.6             |                   |                       |          | 18.2              |                  | 26.8     |

**Table S4 (continued)**

Figure 5 in the revised manuscript: The interannual variability of N and S deposition, emissions and component proportion in China from 2005 to 2020. The emission data over China were taken from MEIC.

---

## Author Comment (AC4)

**SUPPLEMENT FOR THE RESPONSE TO REVIEWER #3**

**Number of tables:1    Number of figures: 6**

**Table list**

Table S9 in the revised supplement: Comparison of the annual $V_d$ of nitrogen compounds by land use type in this and other studies (cm s$^{-1}$).

**Figure list**

Figure R1 The relative importance of interpretation variables (RIV) to the RF predictions. See Table S2 for the meanings of the abbreviations in the figure.

Figure R2. (a-c) Annual mean wet and dry deposition of the N species derived from two complementary databases: (1) published data of bulk N deposition covering 1980–2018 and (2) wet and dry deposition based on the machine learning methods for 2005-2020 in this work. For (1), the blue open circles represent annual average bulk N deposition. The orange curve shows the trends in inorganic N bulk deposition, while the orange dots represent the 5-year average bulk deposition. For (2), the grey dots represent the dry deposition and the yellow dots represent the bulk deposition. (d) The interannual variations of satellite-derived VCDs and total deposition for China in 2005-2020. All the data are relative to the 2005 levels

Figure 2 in the revised manuscript: Methodology framework to estimate dry and wet deposition of this study. The blue process shows the four steps to establish the RF model. The orange process shows the three steps in establishing a GAM model. See Sections 2.2 to 2.3 of the method section in the text for the acquisition of the preliminary data set.

Figure S1 in the revised Supplement: Correlations between simulated $SO_4^{2-}$ and $SO_2$ concentrations from GAM.

Figure S2 in the revised supplement (Figure S1 in the original submission): The RF algorithm monthly performance of CNEMC with the 10-fold cross validation. $R^2$ and RMSE are calculated with equations below the figure (the unit of RMSE are kg N/S ha$^{-1}$ yr$^{-1}$).

Figure S3 in the revised supplement (Figure S2 in the original submission): The same as Figure S2 but for NNDMN.

**Table S9 in the revised supplement: Comparison of the annual $V_d$ of nitrogen compounds by land use type in this and other studies (cm s$^{-1}$).**

| Land use type | Deposition velocity (cm s$^{-1}$) | | | | | | References |
|---|---|---|---|---|---|---|---|
| | NO$_2$ | HNO$_3$ | NO$_3^-$ | NH$_3$ | NH$_4^+$ | SO$_2$ | |
| Farmland | 0.17 | 1.45 | 0.15 | 0.43 | 0.15 | 0.44 | This study |
| | 0.18 | 1.52 | 0.19 | 0.40 | 0.19 | | Xu et al. (2015) |
| | 0.10 | 0.76 | 0.25 | 0.18 | | 0.25 | Zhang et al. (2004) |
| | | | | | | 0.56 | Zhang et al. (2003) |
| | | | | | | | |
| Urban | 0.14 | 1.37 | 0.15 | 0.44 | 0.15 | 0.44 | This study |
| | 0.06 | | | 0.78 | | | Pan et al. (2012) |
| | 0.03 | | | | | 0.20 | Su et al. (2012) |
| | | | | | | 0.55 | Zhang et al. (2003) |
| | 0.07 | 1.77 | 0.44 | 0.28 | 0.44 | | Li et al. (2013) |
| | 0.30 | 1.10 | 0.24 | 0.50 | 0.24 | | Luo et al. (2013) |
| | | | | | | | |
| Coastal | 0.16 | 1.56 | 0.10 | 0.65 | 0.13 | 0.66 | This study |
| | 0.01 | 0.63 | | 0.63 | | | Zhang et al. (2010) |
| | 0.01 | 0.84 | 0.27 | 0.55 | 0.27 | 0.63 | Zhang et al. (2004) |
| | | | | | | 0.40 | Su et al. (2012) |
| | | | | | | | |
| Forest | 0.19 | 2.23 | 0.16 | 0.41 | 0.16 | 0.46 | This study |
| | 0.10 | 2.45 | 0.30 | 0.20 | 0.30 | | Zhang et al. (2004) |
| | 0.19 | 2.23 | 0.16 | 0.41 | 0.16 | | Xu et al. (2015) |
| | 0.04 | | | | | 0.16 | Su et al. (2012) |
| | | | | | | | |
| Grassland | 0.15 | 1.09 | 0.19 | 0.38 | 0.19 | 0.33 | This study |
| | 0.13 | 1.16 | 0.28 | 0.23 | 0.28 | 0.37 | Zhang et al. (2004) |
| | 0.15 | | | | | | Xu et al. (2015) |
| | | | | | | 0.49 | Zhang et al. (2003) |

Note: Zhang et al. (2004), Su et al. (2012), Xu et al. (2015), Zhang et al. (2010) and Zhang et al. (2003) applied RegADMS, NAQPMS, GEOS-Chem, MM5/CMAQ and AURAMS, respectively. In particular, Zhang et al (2003) focused on the global land use and did not provide specific discussion for China, and was thus excluded when calculating the mean of China.

**Figure R1 The relative importance of interpretation variables (RIV) to the RF predictions. See Table S2 for the meanings of the abbreviations in the figure.**

[Figure]

**Figure R2 (a-c) Annual mean wet and dry deposition of the N species derived from two complementary databases: (1) published data of bulk N deposition covering 1980–2018 and (2) wet and dry deposition based on the machine learning methods for 2005-2020 in this work. For (1), the blue open circles represent annual average bulk N deposition. The orange curve shows the trends in inorganic N bulk deposition, while the orange dots represent the 5-year average bulk deposition. For (2), the grey dots represent the dry deposition and the yellow dots represent the bulk deposition. (d) The interannual variations of satellite-derived VCDs and total deposition for China in 2005-2020. All the data are relative to the 2005 levels.**

[Figure]

**Figure 2 in the revised manuscript: Methodology framework to estimate dry and wet deposition of this study. The blue process shows the four steps to establish the RF model. The orange process shows the three steps in establishing a GAM model. See Sections 2.2 to 2.3 of the method section in the text for the acquisition of the preliminary data set.**

[Figure]

**Figure S1 in the revised Supplement: Correlations between simulated $SO_4^{2-}$ and $SO_2$ concentrations from GAM.**

[Figure]

**Figure S2 in the revised supplement (Figure S1 in the original submission): The RF algorithm monthly performance of CNEMC with the 10-fold cross validation. $R^2$ and RMSE are calculated with equations below the figure (the unit of RMSE are kg N/S ha$^{-1}$ yr$^{-1}$).**

[Figure]

Note: The $R^2$, RMSE, MPE and RPE were calculated using following equations ($P$ and $O$ indicates the results from prediction and observation, respectively):

$$R^2 = \frac{\sum_{i=1}^{n}(P_i - \overline{O})^2}{\sum_{i=1}^{n}(O_i - \overline{O})^2}$$

$$RMSE = \sqrt{\frac{1}{n}\sum_{i=1}^{n}(P_i - O_i)^2}$$

**Figure S3 in the revised supplement (Figure S2 in the original submission): The same as Figure S2 but for NNDMN.**

[Figure]

**References**

Pan, Y. P., Wang, Y. S., Tang, G. Q., and Wu, D.: Wet and dry deposition of atmospheric nitrogen at ten sites in Northern China, Atmos. Chem. Phys., 12, 6515-6535, https://doi.org/10.5194/acp-12-6515-2012, 2012.

Su, H., Yin, Y., Zhu, B., Wang, Z., Li, J., and Pan, X.: Numerical simulation and sensitive factors analyse for dry deposition of $SO_2$ and $NO_2$ in Bohai Rim area of China, China Environ. Sci., 32, 1921-1932, 2012.

Xu, W., Luo, X. S., Pan, Y. P., Zhang, L., Tang, A. H., Shen, J. L., Zhang, Y., Li, K. H., Wu, Q. H., Yang, D. W., Zhang, Y. Y., Xue, J., Li, W. Q., Li, Q. Q., Tang, L., Lu, S. H., Liang, T., Tong, Y. A., Liu, P., Zhang, Q., Xiong, Z. Q., Shi, X. J., Wu, L. H., Shi, W. Q., Tian, K., Zhong, X. H., Shi, K., Tang, Q. Y., Zhang, L. J., Huang, J. L., He, C. E., Kuang, F. H., Zhu, B., Liu, H., Jin, X., Xin, Y. J., Shi, X. K., Du, E. Z., Dore, A. J., Tang, S., Collett, J. L., Goulding, K., Sun, Y. X., Ren, J., Zhang, F. S., and Liu, X. J.: Quantifying atmospheric nitrogen deposition through a nationwide monitoring network across China, Atmos. Chem. Phys., 15, 12345-12360, https://doi.org/10.5194/acp-15-12345-2015, 2015.

Zhang, L., Brook, J. R., and Vet, R.: Evaluation of a non-stomatal resistance parameterization for $SO_2$ dry deposition, Atmos. Environ., 37, 2941-2947, https://doi.org/10.1016/s1352-2310(03)00268-1, 2003.

Zhang, Y., Wang, T. J., Hu, Z. Y., and Xu, C. K.: Temporal variety and spatial distribution of dry deposition velocities of typical air pollutants over different land-use types, Climatic Environ. Res., 9, 591-604, 2004. (In Chinese)

---

## Author Response (AR1)

**Response to reviewers' comments and main revisions**

Journal: Atmospheric Chemistry and Physics

Manuscript No.: egusphere-2023-620

Title: Estimating nitrogen and sulfur deposition across China during 2005-2020 based on multiple statistical models

Authors: Kaiyue Zhou, Wen Xu, Lin Zhang, Mingrui Ma, Xuejun Liu, Yu Zhao

We thank very much for the valuable comments and suggestions from the editor and reviewers, which help us improve our manuscript. The comments have been carefully considered and revisions have been made in response to suggestions. Following are our point-by-point responses to the comments and corresponding revisions. Please note that the line/table/figure numbers mentioned following refer to the clean version of the revised manuscript, unless specifically noted.

**Comments from Reviewer #1**

Q1. This study aims to develop a machine learning framework for estimating spatial distribution and long-term trend of N and S deposition across China. Estimated dataset during the period from 2005 to 2020 is valuable to understand effects of emission reductions on deposition and N and S input to ecosystems in China. On the other hand, the dataset has considerable uncertainties (as the authors mentioned in section 3.4). The authors should also take the uncertainties into account in other sections. In addition, there are some parts where discussion is insufficient.

**Response and main revisions:**

We appreciate the reviewer's positive remarks and important comments of this work. Briefly, in the revised manuscript, we have expanded uncertainty analysis and discussion on the overestimation of bulk deposition (please see our response to Question 3 of reviewer, as well as that to Question 2 of Reviewer #3), and transformation of  $SO_2$  to sulfate (please also see our response to Questions 7 and 28 of Reviewer#2). We have also added necessary discussions in "Materials and methods" and "Results and discussions" sections, based on the reviewer's valuable comments. Please find the point-by-point response and corresponding revisions below.

Q2. To estimate dry deposition flux, deposition velocity ( $V_d$ ) was calculated by CTM (GEOS-Chem). Current  $V_d$  models (resistant models) have large uncertainties, especially for gaseous and particulate Nr components. Therefore, the authors should open  $V_d$  calculation in detail. Although the authors indicate gaseous  $V_d$  parameterization in L255 (Wesely, 1989) used in this study, aerosol  $V_d$ parameterization should be indicated too. General aerosol models output  $V_d$  by size. On the other hand, monitored particulate  $NH_4^+$ ,  $NO_3^-$  do not have detailed size information (only the information of cutoff size:  $PM_{2.5}$ ,  $PM_{10}$ , or TPM). It is necessary to explain how to treat the aerosol size to calculate the dry deposition based on equation (1). Moreover, calculated  $V_d$  values should be indicated. For example, average values of  $V_d$  for each land use are very informative for relevant researchers. This will be important information when comparing the dataset with the results of other studies.

**Response and main revisions:**

We thank and agree the reviewer's important comment. As summarized below, we have included the calculation process for the dry deposition velocities ( $V_d$ ), average  $V_d$  values by land use type, and the consistency between modeling and monitoring in particle size in the revised manuscript and supplement.

Firstly, we have expanded the calculation process of  $V_d$ . The process follows a standard big-leaf resistance-in-series model as described by Wesely (1989) for gases and Zhang et al. (2001) for aerosol. The equations and principles have been added in the Text Section in the revised supplement. In particular, the aerodynamic resistance to turbulent transfer from the measurement heights (~3 m) to the roughness

height is estimated using the MERRA-2 data. The surface resistance is calculated based on the Global Land Cover Characteristics Data Base Version 2.0 (http://edc2.usgs.gov/glcc/globdoc2\_0.php), which defines land types (e.g., urban, forest, etc.) at 1 km  $\times$  1 km resolution and is then binned to the model resolution as fraction of the grid cell covered by each land type. Bi-directional NH3 exchange is not considered in the model.

Secondly, monthly  $V_d$  was obtained as the average of hourly values for further estimation of dry deposition flux of N and S species. The annual averages of  $V_d$  have been provided by land use type in a newly added Table S3 in the revised supplement.

Finally, the consistency in particle size between monitored concentration and modeled  $V_d$  has been discussed. In this study, the concentrations of NH4+ and NO3- aerosols were measured using the DELTA active sampling system (DEnuder for Long-Term Atmospheric sampling; described in detail in Flechard et al., 2011). Briefly, the sampling train consists of four denuders and two filters to collect gaseous and particulate N species, respectively. This series does not apply staged cut-off instruments for aerosol sampling, and the empirically determined effective size cut-off is of the order of 4.5 µm, without particle size distribution (Lines 226-230 in the revised manuscript). As NH4+ and NO3- are mainly distributed in the fine particle mode, the samples collected by the DELTA system are considered to represent the content of total particulate matter. Correspondingly, the  $V_d$  simulated with GEOS-Chem indicates the total particles as well. The dry deposition flux of particulate NH4+ and NO3- was then calculated by multiplying the measured concentrations with simulated  $V_d$ .

| Land use categories       | HNO 3 | NH3  | NH 4 | NO 2 | NO 3 | SO 2 | SO 4 |
|---------------------------|------------------|------|-----------------|-----------------|-----------------|-----------------|-----------------|
| Paddy fields              | 1.63             | 0.47 | 0.14            | 0.20            | 0.14            | 0.45            | 0.14            |
| Dry land                  | 1.42             | 0.42 | 0.16            | 0.17            | 0.16            | 0.41            | 0.16            |
| Forestland                | 2.55             | 0.49 | 0.14            | 0.24            | 0.14            | 0.46            | 0.14            |
| Shrub forest              | 1.83             | 0.45 | 0.16            | 0.21            | 0.16            | 0.43            | 0.16            |
| Sparse forestland         | 1.96             | 0.47 | 0.15            | 0.22            | 0.15            | 0.44            | 0.15            |
| Other forestland          | 2.17             | 0.53 | 0.14            | 0.22            | 0.14            | 0.52            | 0.14            |
| High coverage grassland   | 1.29             | 0.36 | 0.18            | 0.12            | 0.18            | 0.36            | 0.18            |
| Medium coverage grassland | 1.05             | 0.34 | 0.18            | 0.09            | 0.18            | 0.34            | 0.18            |
| Low coverage grassland    | 0.88             | 0.31 | 0.17            | 0.04            | 0.17            | 0.31            | 0.17            |
| River channel             | 1.17             | 0.38 | 0.15            | 0.13            | 0.15            | 0.37            | 0.15            |
| Lakes                     | 0.93             | 0.32 | 0.19            | 0.07            | 0.19            | 0.32            | 0.19            |
| Reservoir pond            | 1.37             | 0.43 | 0.14            | 0.14            | 0.14            | 0.43            | 0.14            |
| Permanent glacial snow    | 0.59             | 0.27 | 0.16            | 0.03            | 0.16            | 0.28            | 0.16            |
| Tidal-flat                | 1.06             | 1.01 | 0.07            | 0.02            | 0.07            | 1.02            | 0.07            |
| Beach land                | 1.30             | 0.37 | 0.16            | 0.12            | 0.16            | 0.36            | 0.16            |
| Urban land use            | 1.37             | 0.44 | 0.15            | 0.14            | 0.15            | 0.44            | 0.15            |
| Rural settlements         | 1.27             | 0.40 | 0.15            | 0.14            | 0.15            | 0.40            | 0.15            |
| Other construction land   | 1.40             | 0.47 | 0.14            | 0.16            | 0.14            | 0.47            | 0.14            |
| Sand                      | 0.87             | 0.30 | 0.10            | 0.02            | 0.10            | 0.30            | 0.10            |
| Gobi                      | 0.98             | 0.30 | 0.10            | 0.02            | 0.10            | 0.30            | 0.10            |
| Saline alkali soil        | 0.87             | 0.31 | 0.15            | 0.04            | 0.15            | 0.31            | 0.15            |
| Swamp land                | 1.61             | 0.38 | 0.15            | 0.14            | 0.15            | 0.37            | 0.15            |
| Bare land                 | 1.10             | 0.30 | 0.10            | 0.03            | 0.10            | 0.30            | 0.10            |
| Bare rock                 | 0.88             | 0.30 | 0.14            | 0.03            | 0.14            | 0.30            | 0.14            |
| Other unused land         | 0.72             | 0.29 | 0.18            | 0.04            | 0.18            | 0.30            | 0.18            |

Table S3 in the revised supplement: The modeled  $V_d$  for different land use categories (cm s-1).

Note: Land-Use and Land-Cover Change (LUCC) data were obtained from Resource and Environment Data Cloud Platform (http://www.resdc.cn/), generated by manual visual interpretation of Landsat TM/ETM remote sensing images.

Q3. This study uses wet deposition of  $SO_4^{2-}$  (EANET) and wet or bulk deposition of  $NO_3^-$ ,  $NH_4^+$  (NNDMN). There is a need to discuss which regions the overestimation of  $NO_3^-$ ,  $NH_4^+$  by bulk sampling may affect in "3 Results and discussion".

**Response and main revisions:**

We thank and agree the reviewer's important comment. We have expanded the discussion on the uncertainty from bulk deposition sampling **in Lines 697-705 in the revised manuscript**. The uncertainty was greater in areas with a higher proportion of dry to total deposition (such as NW and NE areas with less precipitation), and smaller in areas with a lower proportion (such as SE and SW with more precipitation). For example, we compared the results from Kuang et al. (2016) and Song et al. (2017), and found that the difference between 5-year in situ measurements of bulk and wet dissolved inorganic nitrogen (DIN) deposition was only about 2.5 kg N ha-1 yr-1 at a rural site in SW, equaling to 12% of annual bulk deposition. As SE is the most developed region in China, with relatively high emissions and deposition across the country, the uncertainty from bulk deposition measurement and application is of limited impact on the national level or the overall spatial pattern of deposition.

Q4. L180: Does "chemical transport mode (CTM) results" means emission inventory? If so, "emission inventory" should be used as shown in Fig. 2. "mode" may be mistake for "model".

**Response and main revisions:**

We appreciate the reviewer's reminder. "**Chemical transport model (CTM) results**" means surface concentrations of  $NO_3^-$ ,  $HNO_3$ ,  $NH_4^+$ , and  $SO_4^{2-}$  simulated from CTM simulation. Emission inventories were also included in RF. We have accordingly modified **Figure 2 in the revised manuscript**.

Q5. L184: "dry deposition rate" means dry deposition amount. "dry deposition velocity" is correct.

**Response and main revisions:**

We thank the reviewer's reminder and "dry deposition rate" has been modified as "dry deposition velocity" **in Lines 123 and 185 in the revised manuscript**.

Q6. L215: Regarding particulate  $NH_4$ ,  $NO_3$ , particle size information should be indicated (TPM,  $PM_{10}$ ,  $PM_{2.5}$  etc.). In the case of  $PM_{2.5}$ , dry deposition of  $NO_3$  in coarse aerosols, which contributes considerable part of total  $NO_3$  dry deposition, is ignored. If so, it should be discussed in section 3.4.

**Response and main revisions:**

We thank the reviewer's important comment. The dry deposition flux of  $NH_4^+$ and  $NO_3^-$  was calculated by multiplying measured concentrations with simulated  $V_d$ from the GEOS-Chem. Both the observed concentrations and simulated  $V_d$ represented (or approximated) the values for total particles (Please also see our response to Question 2).

Q7. L406-410: Trend analysis (e.g. Mann-Kendall test) is effective to mention statistically that  $R_{dry/wet}$  is stable for N, decline for S before 2015. It is also available for the increases after 2015.

**Response and revisions:**

We thank the reviewer's important comment. Following his/her suggestion, we conducted Mann-Kendall test for the trend of  $R_{dry/wet}$  for N and S, and the results are presented **in a newly added Table S5 in the revised supplement**. According to the test, the  $R_{dry/wet}$  of N species kept relatively stable for earlier years and then slightly increased since 2015 (p>0.01). The  $R_{dry/wet}$  of sulfur declined significantly before 2015 (p>0.01) and then slightly increased afterwards without statistical significance (p>0.01). Therefore, our statement was supported by the test.

Table S5 in the revised supplement: The Mann-Kendall test for the trend of  $R_{dry/wet}$  of N and S in 2005-2020. The z-value represents the standard normal statistic, and the p-value represents the generalization. The former indicates the trend, while the latter indicates statistical significance. P1 and P2 indicate 2005–2015 and 2015–2020, respectively.

| Species | OX     | KN    | RE     | DN    | Ν      |       | S      |       |
|---------|--------|-------|--------|-------|--------|-------|--------|-------|
| Period  | P1     | P2    | P1     | P2    | P1     | P2    | P1     | P2    |
| Z       | -2.024 | 2.254 | -2.024 | 1.879 | -2.336 | 1.879 | -3.270 | 1.127 |
| р       | 0.043  | 0.024 | 0.043  | 0.060 | 0.020  | 0.060 | 0.001  | 0.260 |

Note: Negative and positive z-value indicate a downward and upward trend in the time series, respectively; p<0.01 indicates a significance level of 99%.

Q8. L420-423: In Figure S3, the range of temperature variation during the period is within 1 K. I think it is too small to enhance dry deposition by stomatal uptake. Moreover, dry deposition of these N and S component (aerosols and reactive gases) to stomata is small compared to deposition to cuticle.

**Response and revisions:**

We thank and agree the reviewer's comment. The temperature variation is too small to enhance dry deposition, thus we have deleted the sentence and modified **Figure S4 in the revised supplement (Figure S3 in the original submission)**.

Q9. L440-445: Please check the amount of N deposition in USA and Europe in Table S4. They are too small compared with the global distribution of total N deposition (Fig. 4.8a) by Vet et al (2014) in L883. It shows the range of 1-20 kg N and 2-40 kg N in USA and Europe, respectively. Schwede and Lear (2014) also shows same range of N deposition in USA. (http://dx.doi.org/10.1016/j.atmosenv.2014.04.008)

**Response and revisions:**

We thank the reviewer's reminder and we are sorry for the error in the calculation in our original submission. We have checked and corrected the results in **Table S6 in the revised supplement (Table S4 in the original submission)**. The corrected results are within the range provided by Vet et al (2014), as shown in Table R1.

Table S6 in the revised supplement: Comparisons of total deposition fluxes of different species between our study in China and two networks in other countries (kg N/S ha-1 yr-1).

|        | Period    | RDN  | OXN  | Ν    | S    |
|--------|-----------|------|------|------|------|
| USA    | 1990-2020 | 2.7  | 13.3 | 16.0 | 16.1 |
| Europe | 2000-2019 | 1.4  | 4.7  | 6.1  | 4.1  |
| China  | 2005-2020 | 20.2 | 15.2 | 35.4 | 25.9 |

Table R1 Comparisons of total N deposition fluxes between our study and Vet et al (2014) (kg N/S ha-1 yr-1).

|        | Reference        | Ν        | S        |
|--------|------------------|----------|----------|
| USA    | This study       | 16.0     | 16.1     |
|        | Vet et al (2014) | 0.4-20.0 | 4.0-23.4 |
| Europe | This study       | 6.1      | 4.1      |
|        | Vet et al (2014) | 2.0-28.1 | 4.0-32.0 |

Q10. L471-485: The authors should discuss why the short-term emission reduction was not well reflected in the deposition. For example, Yamaga et al. (2021) in L935 mentioned that recent decrease of total S deposition in Japan was associated with recent reduction in  $SO_2$  in China. Therefore, the short-term emission reduction might be reflected in decrease of transboundary air pollution at first, because the reduction started from the east side in China.

**Response and revisions:**

We thank and agree the reviewer's very important comment. It is really an impressive thought (also raised by Reviewer #2, see his Questions 19 and 25). As reported by Yamaga et al. (2021), the trend of  $NO_3^-$  to nss- $SO_4^{2^-}$  concentration ratio in precipitation in Japan clearly corresponded to that of the  $NO_X$  to  $SO_2$  emission ratio in China, indicating that the OXN and S deposition in Japan might be influenced by the  $NO_X$  and  $SO_2$  emissions in China, respectively. Therefore, the short-term emission reduction in China was likely to reduce the transboundary deposition to downwind areas (such as Japan). Such transboundary impact might be sooner than the local one. We have stated in Lines 523-527 in the revised manuscript.

**Q11. L600-607: The authors are requested to discuss why the larger N deposition was found in summer.**

**Response and revisions:**

We thank the reviewer's reminder and have added the reasons why the larger N deposition was found in summer in Lines 657-660 in the revised manuscript. The seasonal trend of N deposition was largely influenced by dry deposition, given its large proportion to the total. As shown in a newly added Figure S5 in the revised supplement, the  $V_d$  of HNO3 in summer was 4.4 times in winter, leading to larger OXN deposition in summer. Moreover, warm weather elevated the volatility of NH3 in croplands, resulting in greater emissions and thereby deposition in summer.

Figure S5 in the revised supplement: The monthly means of the modeled dry deposition velocity of N and S during 2013-2020.

**Reference**

Flechard, C. R., Nemitz, E., Smith, R. I., Fowler, D., Vermeulen, A. T., Bleeker, A., Erisman, J. W., Simpson, D., Zhang, L., Tang, Y. S., and Sutton, M. A.: Dry deposition of reactive nitrogen to European ecosystems: a comparison of inferential models across the NitroEurope network, Atmos. Chem. Phys., 11, 2703-2728, https://doi.org/10.5194/acp-11-2703-2011, 2011.

Kuang, F., Liu, X., Zhu, B., Shen, J., Pan, Y., Su, M., and Goulding, K.: Wet and dry nitrogen deposition in the central Sichuan Basin of China, Atmos. Environ., 143, 39-50, https://doi.org/10.1016/j.atmosenv.2016.08.032, 2016.

Song, L., Kuang, F., Skiba, U., Zhu, B., Liu, X., Levy, P., Dore, A., and Fowler, D.: Bulk deposition of organic and inorganic nitrogen in southwest China from 2008 to 2013, Environ. Pollut., 227, 157-166, https://doi.org/10.1016/j.envpol.2017.04.031, 2017. Vet, R., Artz, R. S., Carou, S., Shaw, M., Ro, C.-U., Aas, W., Baker, A., Bowersox, V. C., Dentener, F., Galy-Lacaux, C., Hou, A., Pienaar, J. J., Gillett, R., Forti, M. C., Gromov, S., Hara, H., Khodzher, T., Mahowald, N. M., Nickovic, S., Rao, P. S. P., and Reid, N. W.: A global assessment of precipitation chemistry and deposition of sulfur, nitrogen, sea salt, base cations, organic acids, acidity and pH, and phosphorus, Atmos. Environ., 93, 3-100, https://doi.org/10.1016/j.atmosenv.2013.10.060, 2014.

Wesely, M. L.: Parameterization of surface resistances to gaseous dry deposition in regional-scale numerical models, Atmos. Environ., 23, 1293-1304, https://doi.org/https://doi.org/10.1016/0004-6981(89)90153-4, 1989.

Yamaga, S., Ban, S., Xu, M., Sakurai, T., Itahashi, S., and Matsuda, K.: Trends of sulfur and nitrogen deposition from 2003 to 2017 in Japanese remote areas, Environ. Pollut., 289, 117842, https://doi.org/10.1016/j.envpol.2021.117842, 2021.

Zhang, L., Gong, S., Padro, J., and Barrie, L.: A size-segregated particle dry deposition scheme for an atmospheric aerosol module, Atmos. Environ., 35, 549-560, https://doi.org/10.1016/S1352-2310(00)00326-5, 2001.

**Comments from Reviewer #2**

Q1. It is of great interest and importance to modeling the historical S/N deposition in China, one of the hotspots in the world, for supporting policy-making. Based on various databases and applying machine learning method, the authors estimate the deposition of different species of N and S at relatively high resolution during 2005-2020. A delayed response of N/S deposition to NOX/SO2 emission abatement was found in China. In general, the manuscript was well written, and worth publishing after some minor revision. The reason of delay need be further discussion, e.g., the transport of N/S out of land of China and the changes of atmospheric oxidizing capacity.

**Response and main revisions:**

We appreciate the reviewer's positive comment, and have made point-by-point response and revisions as summarized below. In particular, the delayed response of deposition to emission control is of great interest to research community and more careful studies are needed to answer the question. In this work, we have added discussions on the transport out of China and changing atmospheric oxidizing capacity. Please find our response to Questions 19 and 25, and to Questions 7 and 28, respectively.

**Q2. Line 30: The full name of OXN or RDN need be given when first occurs. Line 34 & 35: Same as above for $R_{dry/wet}$ and $R_{RDN/OXN}$ .**

**Response and main revisions:**

We thank the reviewer's reminder and have added the full name of OXN, RDN,  $R_{dry/wet}$  and  $R_{RDN/OXN}$  in Lines 30-31, 34, and 36-37 in the revised manuscript.

**Q3. Line 68: to reduce acid rain and later improve ...**

**Response and main revisions:**

We thank the reviewer's reminder and the sentence has been modified as suggested in Lines 70-72 in the revised manuscript.

**Q4. Line 72: Total emission control of NOX was carried out in the 12th FYP.**

**Response and main revisions:**

We thank the reviewer's reminder and we have corrected this error as "including the policy of limiting national total emission levels of  $SO_2$  and  $NO_X$  within the 11th Five-year Plan (FYP, 2006-2010) and 12th FYP period (2011-2015) respectively" in Lines 73-75 in the revised manuscript.

**Q5. Line 77: Which years?**

**Response and main revisions:**

We thank the reviewer's reminder and the sentence has been modified as ".....those policies have reduced annual  $SO_2$  and  $NO_X$  emissions from 2007 and 2012, respectively." in Line 79 in the revised manuscript.

**Q6. Line 138: (GAM)**

**Response and main revisions:**

We thank the reviewer's reminder and have modified as "(GAM)" in Line 140 in the revised manuscript.

Q7. Line 223: The transformation of  $SO_2$  to sulfate depends on atmospheric oxidizing capacity, thus on  $NO_X$  concentration (or emission). The discussion on the uncertainty need added.

**Response and main revisions:**

We thank the reviewer's important comment and have added the discussion on the uncertainty in Lines 684-691 in the revised manuscript.  $SO_2$  can be oxidized to form sulfate in the troposphere, which may occur in the gas phase, clouds or fog

13

droplets, or on aerosol particles (Lee et al., 2019; Sarwar et al., 2013). Those processes are influenced by the atmospheric oxidizing capacity, and thereby the NOX concentration (He et al., 2014; Ye et al., 2023). Along with economic development and implementation of air pollution controls, the changing emissions of NOX as well as some other species (e.g., volatile organic compounds) have altered atmospheric oxidizing capacity within the research period. Using the relationship between 2013 and 2020 to extrapolate the sulfate deposition for 2005-2020 would potentially result in some uncertainty to the results.

**Q8. Line 257: What were the modeled years? How was the performance of the model for China.**

**Response and main revisions:**

We are sorry for the ambiguity and thank the reviewer's reminder. The simulation was conducted for 2005-2020 with good model performance. The correlation coefficients between simulated and observed concentrations ranged 0.51–0.82 and the normalized mean biases were within 30%. We have added the model performance and the relevant reference in Lines 285-288 in the revised manuscript.

**Q9.** Line 299: Although there is reference, the brief introduction on the method is needed. What was the performance of the model?**

**Response and main revisions:**

We thank the reviewer's reminder. We have added brief principle and model performance of GAM in the revised manuscript. The GAM model connects the nonlinear relationship between wet deposition and predicted variables (satellite-derived VCDs, meteorological factors and geographic covariates, etc.) to estimate the monthly wet deposition of  $SO_4^{2^-}$ ,  $NO_3^-$  and  $NH_4^+$  in China at a horizontal resolution of  $0.25^{\circ} \times 0.25^{\circ}$  (Lines 351-355 in the revised manuscript). The predicted wet deposition was basically consistent with the ground-level observed values, and

the correlation coefficient was greater than 0.7, indicating the good performance of GAM (Lines 355-359 in the revised manuscript).

Q10. Line 345: Were these studies carried out for the whole country or just at several sites? Since the monitoring sites are concentrated in the more developed east part of China, how did you consider the uncertainty caused by the bias?

**Response and main revisions:**

We thank the reviewer's comment and have added the research scale information in the Table S4 in the revised manuscript (Table S3 in the original submission). Indeed the studies conducted in less developed west China were much less than those in east. As those studies were not included in the RF or GAM, the uneven distribution of monitoring sites would not directly influence our prediction results, but might bring bias to the comparison between prediction and observation. The bias is expected to be further evaluated and reduced when more observation data get available in the west.

**Q11. Line 353: in its high stage?**

**Response and main revisions:**

We thank the reviewer's reminder. We mean the control of  $SO_2$  started from 2005, thus it should be "in its initial stage" in Line 390 in the revised manuscript, not in its high stage.

**Q12. Line 378: Limited before 2010. The total emission control of $NO_X$ was carried out in the 12th FYP (2011-2015).**

**Response and main revisions:**

We thank the reviewer's reminder and the sentence has been modified as "China's  $NO_X$  emission control was limited till 2010" in Lines 414-415 in the revised manuscript.

Q13. Line 379: The total emission control of  $NO_X$  in the 12th FYP required the installation of SCR since 2011, although more and more SCR installation had been finished after 2013.

**Response and main revisions:**

We thank the reviewer's reminder and the sentence has been modified as "The country required installation of selective catalyst reduction (SCR) systems from 2011, and NAPPCAP drove fast growing penetration of SCR in the power and cement production sectors, resulting in a 28.6% reduction in the annual total emissions of NOX from 2013 to 2020 (Karplus et al., 2018; Li et al., 2018)" in Lines 416-419 in the revised manuscript.

**Q14. Line 418: Another reason that increasing $NO_X$ led to high atmospheric oxidizing capacity, and thus promoting sulfate formation for wet deposition?**

**Response and main revisions:**

We thank and agree the reviewer's important comment. The transformation of  $SO_2$  to sulfate depends on atmospheric oxidizing capacity and thereby the  $NO_X$  concentration (Please also see our response to Question 7). The growth of  $NO_X$  emissions before 2013 led to high atmospheric oxidizing capacity, thereby promoting sulfate formation for wet deposition. We have added the discussion in Lines 459-461 in the revised manuscript.

**Q15. Line 421: How about the decrease in atmospheric oxidizing capacity?**

**Response and main revisions:**

We thank the reviewer's comment. Although  $NO_X$  was declining for most recent years, the atmospheric oxidizing capacity was still growing with enhanced ground-level ozone concentration in most areas of China (Feng et al., 2021; Wang et al., 2023). The reasons include less effective control on the emissions of volatile organic compounds (Ding et al., 2021; Zheng et al., 2018). Therefore we mean "decrease in atmospheric oxidizing capacity" is not a strong explanation for the growing  $R_{dry/wet}$ .

**Q16. Line 453: Replacing continents by regions? Same for the whole text.**

**Response and main revisions:**

We thank the reviewer's reminder and the "continents" has been modified as "regions" in Lines 493, 512-513 in the revised manuscript.

**Q17. Line 463: Delete 'in'. Need more recent literature on the trends.**

**Response and main revisions:**

We thank and agree the reviewer's comment. We have deleted "in" and included more recent studies (Constantin et al., 2020; Fowler et al., 2013; Skyllakou et al., 2021; Zhao and Qiao, 2022) in Lines 502-504 in the revised manuscript.

**Q18. Line 469: 2005 was only for SO2. NOx had later year.**

**Response and main revisions:**

We thank the reviewer's reminder and the sentence has been modified as "Following developed countries, gradually tightened measures of reducing  $SO_2$  and  $NO_X$  have been launched since 2005 and 2011 respectively, and the deposition began to decline afterwards" in Lines 508-510 in the revised manuscript.

**Q19. Line 485: What were the reasons for the delay? How about the contribution of natural sources for NOx, or the transboundary transport of S/N?**

**Response and main revisions:**

We thank the reviewer's important comment and have made response and corresponding revisions regarding the two issues.

(1) The transport of deposition to downwind areas could be part of reasons for the delay, and we have added the discussions in Lines 523-529 in the revised manuscript. As reported by Yamaga et al. (2021), the trend of  $NO_3^-$  to  $nss-SO_4^{2^-}$  concentration ratio in precipitation in Japan clearly corresponded to that of the  $NO_X$  to  $SO_2$  emission ratio in China, indicating that the OXN and S deposition in Japan might be influenced by the  $NO_X$  and  $SO_2$  emissions in China, respectively. Therefore, the short-term emission reduction in China was likely to reduce the transboundary deposition to downwind areas (such as Japan). Such transboundary impact might be sooner than the local one.

(2) The soil NOX emissions from both the natural nitrogen pool and fertilizer input are conventionally considered as natural sources, and have not been included in the current design of emission control strategies in China yet (Zhang et al., 2019). The nitrogen inputs to soil lead to soil NOX emissions in China reaching 0.4-1.3 Tg N yr-1, about 12% of the anthropogenic NOX emissions (Lu et al., 2019; Lu et al., 2021). Given the small fraction of natural NOX to the total, they could not be the main reason for the delay of deposition to the control of anthropogenic NOX. With continuous control of anthropogenic emissions in the future, however, the variation of emissions from natural sources might play a more important role on the changing deposition and deserves more attentions. (Lines 529-532 in the revised manuscript).

**Q20. Line 506: What are the countries for example?**

**Response and main revisions:**

We thank and agree the editor's comment and the sentence has been modified as "In many European countries with abundant agricultural activities (such as Netherlands, Germany, Switzerland and France)" in Lines 554-555 in the revised manuscript.

**Q21. Line 508: although not as strong as for NOX.**

**Response and main revisions:**

We thank and agree the reviewer's comment and the phrase has been added in line 557 in the revised manuscript.

Q22. Line 538: Delete 'the pollution was mainly transported by atmospheric turbulence and'.

**Response and main revisions:**

We thank the reviewer's reminder and the phrase has been deleted.

**Q23. Line 546: Not dominate, although share more.**

**Response and main revisions:**

We thank the reviewer's reminder and the sentence has been modified as "while farming and animal husbandry shared more in the economy in the west, leading to substantial NH3 emissions" in Lines 592-594 in the revised manuscript.

Q24. Line 551: Did the results of satellite-derived VCDs show so sharp reduction of vertical column densities? How about the possibility of overestimation of the emission reduction?

**Response and main revisions:**

We thank and agree the reviewer's important comment. While MEIC has been recognized as the best estimation of China's air pollutant emissions and widely applied, the uncertainty of China's emission inventory is always a concern in the research community. Satellite-derived VCDs provide some information on the changing emissions, but cannot provide accurate estimates alone as well. We compared MEIC and estimation with a "top-down" methodology based on satellite observation (Qu et al., 2019), and found that MEIC indeed provided a more optimistic

estimation for China's  $SO_2$  emission reduction. We have added the information in Lines 600-604 in the revised manuscript.

Q25. Line 555: I guess higher ratio of emission from the east than west was transported out of land and deposit on the sea. What is the effect on the effectiveness?

**Response and main revisions:**

We thank and agree the reviewer's important comment. As we respond to Question 19 (and Question 10 of Reviewer #1), the changing transboundary deposition to downwind areas from eastern China could be part of reason for the delayed response of local deposition to changing emissions. We have stressed this in Lines 523-529 in the revised manuscript. Regarding the difference in changing deposition between eastern and western China, the controls of precursor emissions should still be the dominating driving factor. Please also see our response to Question 19.

**Q26. Line 577: The D/E ratio for the whole China need be added in the figure.**

**Response and main revisions:**

We thank and agree the reviewer's important comment. We have added the D/E ratio for the whole China **in Figure 9** and "The nationwide D/E of OXN, RDN, and S were 1.4, 2.4, and 2.3, respectively." **in Lines 629-630** in the revised manuscript.

---

## Author Response (AR2)

**Response to reviewers' comments and main revisions**

Journal: Atmospheric Chemistry and Physics

Manuscript No.: egusphere-2023-620

Title: Estimating nitrogen and sulfur deposition across China during 2005-2020 based on multiple statistical models

Authors: Kaiyue Zhou, Wen Xu, Lin Zhang, Mingrui Ma, Xuejun Liu, Yu Zhao

We thank very much for the comments and suggestions from the reviewer, which help us improve our manuscript. Following are responses to the comments and corresponding revisions. Please note that the line numbers mentioned following refer to the clean version of the revised manuscript, unless specifically noted.

**Comments from Reviewer #2**

The manuscript was well revised according to the reviews' comments. Especially, the delay of deposition decline was discussed. Some very minor comments are: 1. Line 33: delete one 'a' at the end of this line.

**Response and main revisions:**

We thank the reviewer's reminder and it is corrected as suggested.

**2. Line 46: The reason for the different ratio need be added here as conclusion.**

**Response and main revisions:**

We thank the reviewer's comment and the reason is added as: "....., while smaller deposition to emission ratios (D/E) existed in developed eastern China, attributed to more intensive human activities and thereby anthropogenic emissions."

**3. Line 357: What is the different between bulk and wet-only? Some reference?**

**Response and main revisions:**

We thank the reviewer's comment. Actually we include relevant analysis and references in Section 3.4, as indicated in lines 697-705:

For example, dry deposition was observed to account for around 20% of the bulk at three rural stations in the North China Plain, and the contribution could reach 39% in some urban areas (Zhang et al., 2015; Zhang et al., 2008). In contrast, the difference between bulk and wet deposition of dissolved inorganic nitrogen (DIN) was equal to 12% of the bulk in a rural site in SW (Kuang et al., 2016; Song et al., 2017). Basically, the uncertainty was greater in areas with a higher proportion of dry to total deposition (such as NW and NE areas with less precipitation), and smaller in areas with a lower proportion (such as SE with more precipitation).

**Reference:**

- Kuang, F., Liu, X., Zhu, B., Shen, J., Pan, Y., Su, M., and Goulding, K.: Wet and dry nitrogen deposition in the central Sichuan Basin of China, Atmos. Environ., 143, 39-50, https://doi.org/10.1016/j.atmosenv.2016.08.032, 2016.
- Song, L., Kuang, F., Skiba, U., Zhu, B., Liu, X., Levy, P., Dore, A., and Fowler, D.: Bulk deposition of organic and inorganic nitrogen in southwest China from 2008 to 2013, Environ. Pollut., 227, 157-166, https://doi.org/10.1016/j.envpol.2017.04.031, 2017.
- Zhang, G., Pan, Y., Tian, S., Cheng, M., Xie, Y., Wang, H., and Wang, Y.: Limitations of passive sampling technique of rainfall chemistry and wet deposition flux characterization, Res. Environ., 28, 684-690, https://doi.org/10.13198/j.issn.1001-6929.2015.05.03, 2015.
- Zhang, Y., Liu, X. J., Fangmeier, A., Goulding, K. T. W., and Zhang, F. S.: Nitrogen inputs and isotopes in precipitation in the North China Plain, Atmos. Environ., 42, 1436-1448, https://doi.org/10.1016/j.atmosenv.2007.11.002, 2008.

**4. Line 401: Here the national emission of NOx and NH3 should be shown and compared.**

**Response and main revisions:**

We thank the reviewer's important comment. In MEIC, the national emissions of  $NO_X$  and  $NH_3$  were estimated at 7.2 and 8.3 TgN/yr for 2005-2020, respectively. Therefore, the proportions of RDN and OXN to total N deposition (58% and 42%) are close to those of emissions (54% and 46%, respectively). We have added the comparisons in lines 401-403 in the revised manuscript.

**5. Line 403: it fraction in total deposition reaching...**

**Response and main revisions:**

We thank the reviewer's comment and it is revised as suggested.

**6. Line 474: declining R before 2011 resulted...**

**Response and main revisions:**

We thank the reviewer's comment and it is revised as suggested.

**7. Line 574: relatively high NH3...**

**Response and main revisions:**

We thank the reviewer's comment and it is corrected as suggested.

**8. Line 579: The monsoon season is also growing season and fertilization period.**

**Response and main revisions:**

We agree the reviewer's comment, but both dry and wet deposition of RDN could be enhanced in the growing season and fertilization period, thus it cannot be an exclusive reason for the abundant wet deposition.

**9. Line 587: western regions?**

**Response and main revisions:**

We thank the reviewer's reminder and it is corrected as suggested.

**10. Line 607: How about the detailed data on the fluxes out of the land? Did they support for the delay by declining long-range transport?**

**Response and main revisions:**

We thank the reviewer's important comment. Due to lack of observation data, however, we could not quantify the fluxes out of the land with the machine learning techniques in this work. We will take the reviewer's suggestion and continue the research on the S/N transport out of land to further evaluate the delay of deposition decline in the future.